# Stochastic Submodular Bandits with Delayed Composite Anonymous Bandit Feedback

## Abstract

This paper investigates the problem of combinatorial multiarmed bandits with stochastic submodular (in expectation) rewards and full-bandit delayed feedback, where the delayed feedback is assumed to be composite and anonymous. In other words, the delayed feedback is composed of components of rewards from past actions, with unknown division among the sub-components. Three models of delayed feedback: bounded adversarial, stochastic independent, and stochastic conditionally independent are studied, and regret bounds are derived for each of the delay models. Ignoring the problem dependent parameters, we show that regret bound for all the delay models is $\tilde{O}(T^{2/3} + T^{1/3}\nu)$ for time horizon $T$, where $\nu$ is a delay parameter defined differently in the three cases, thus demonstrating an additive term in regret with delay in all the three delay models. The considered algorithm is demonstrated to outperform other full-bandit approaches with delayed composite anonymous feedback. We also demonstrate the generalizability of our analysis of the delayed composite anonymous feedback in combinatorial bandits as long as there exists an algorithm for the offline problem satisfying a certain robustness condition.

## 1 Introduction

Many real world sequential decision problems can be modeled using the framework of stochastic multi-armed bandits (MAB), such as scheduling, assignment problems, ad-campaigns, and product recommendations. In these problems, the decision maker sequentially selects actions and receives stochastic rewards from an unknown distribution. The objective is to maximize the expected cumulative reward over a time horizon. Such problems result in a trade-off between trying actions to learn the system (*exploration*) and taking the action that is empirically the best seen so far (*exploitation*).

Combinatorial MAB (CMAB) involves the problem of finding the best subset of $K$ out of $N$ items to optimize a possibly nonlinear function of reward of each item. Such a problem has applications in cloud storage (Xiang et al., 2014), cross-selling item selection (Wong et al., 2003), social influence maximization (Agarwal et al., 2022), etc. The key challenge in CMAB is the combinatorial $N$-choose-$K$ decision space, which can be very large. This problem can be converted to standard MAB with an exponentially large action space, although needing an exponentially large time horizon to even explore each action once. Thus, the algorithms for CMAB aim to not have this exponential complexity while still providing regret bounds. An important class of combinatorial bandits is submodular bandits; which is based on the intuition that opening additional restaurants in a small market may result in diminishing returns due to market saturation. A set function $f: 2^\Omega \to \mathbb{R}$ defined on a finite ground set $\Omega$ is said to be submodular if it satisfies the diminishing return property: for all $A \subseteq B \subseteq \Omega$, and $x \in \Omega \setminus B$, it holds that $f(A \cup \{x\}) - f(A) \geq f(B \cup \{x\}) - f(B)$ (Nemhauser et al., 1978). Multiple applications for CMABs with submodular rewards have been described in detail in (Nie et al., 2022), including social influence maximization, recommender systems, and crowdsourcing. In these setups, the function is also monotone (adding more restaurants give better returns, adding more seed users give better social influence), where for all $A \subseteq B \subseteq \Omega$, $f(A) \leq f(B)$, and thus we also assume monotononicity in the submodular functions.

Feedback plays an important role in how challenging the CMAB problem is. When the decision maker only observes a (numerical) reward for the action taken, that is known as bandit or full-bandit feedback. When the decision maker observes additional information, such as contributions of each base arm in the action, that is semi-bandit feedback. Semi-bandit feedback greatly facilitates learning. Furthermore, there are two common formalizations depending on the assumed nature of environments: the stochastic setting and the adversarial setting. In the adversarial setting, the reward sequence is generated by an unrestricted adversary, potentially based on the history of decision maker's actions. In the stochastic environment, the reward of each arm is drawn independently from a fixed distribution. For CMAB with submodular and monotone rewards, stochastic setting is not a special case of the adversarial setting since in the adversarial setting, the environment chooses a sequence of monotone and submodular functions $\{f_1, \cdots, f_T\}$, while the stochastic setup assumes $f_t$ to be monotone and submodular in expectation (Nie et al., 2022). In the adversarial setting, even if we limit ourselves to MAB instead of CMAB, the effect of composite anonymous delay appears as a multiplicative factor in the literature (e.g. (Cesa-Bianchi et al., 2018)). In this paper, we study the impact of full-bandit feedback in the stochastic setting for CMAB with submodular rewards and cardinality constraints. In this case, the regret analysis with full-bandit feedback has been studied in the adversarial setting in (Niazadeh et al., 2021), and in stochastic setting in (Nie et al., 2022).

In the prior works on CMAB as mentioned earlier, the feedback is available immediately after the action is taken. However, this may not always be the case. Instead of receiving the reward in a single step, it can be spread over multiple number of time steps after the action was chosen. Following each action choice, the player receives the cumulative rewards from all prior actions whose rewards are due at this specific step. The difficulty of this setting is due to the fact that the agent does not know how this aggregated reward has been constituted from the previous actions chosen. This setting is called delayed composite anonymous feedback. Such feedback arise in multiple practical setups. As an example, we consider a social influence maximization problem. Consider a case of social network where a company developed an application and wants to market it through the network. The best way to do this is selecting a set of highly influential users and hope they can love the application and recommend their friends to use it. Influence maximization is a problem of finding a small subset (seed set) in a network that can achieve maximum influence. This subset selection problem in social networks is commonly modeled as an offline submodular optimization problem (Domingos & Richardson, 2001; Kempe et al., 2003; Chen et al., 2010). However, when the seed set is selected, the propagation of influence from one person to another may incur a certain amount of time delay and is not immediate (Chen et al., 2012). The time-delay phenomena in information diffusion has also been explored in statistical physics (Iribarren & Moro, 2009; Karsai et al., 2011). The spread of influence diffusion, and that at each time we can only observe the aggregate reward limits us to know the composition of the rewards into the different actions in the past. Further, the application developer, in most cases, will only be able to see the aggregate reward leading to this being a bandit feedback. This motivates our study of stochastic CMAB with submodular rewards and delayed composite anonymous bandit feedback.

To the best of our knowledge, this is the first work on stochastic CMAB with delayed composite anonymous feedback. In this paper, we consider three models of delays. The first model of delay is 'Unbounded Stochastic Independent Delay'. In this model, different delay distributions can be chosen at each time, and these delay distributions are independent of each other. The second model is 'Unbounded Stochastic Conditionally Independent Delay'. In this model, the delay distribution does not only depend on time, but also on the set chosen. The third model is 'Bounded Adversarial Delay'. In this model, the maximum delay at each time can be chosen arbitrarily as long as it is bounded. We note that in stochastic cases, the delay is not bounded, while is governed by the tight family of distributions.[1] In the adversarial case, there is a bound on the maximum delay, and the process generating this delay does not need to satisfy any other assumptions. Thus, the results of stochastic and adversarial setups do not follow from each other. In particular, this is the first work where the delay distribution is allowed to change over time. This gives new models for delayed composite anonymous feedback which are more general than that considered in the literature. In each of the three models of delay, this paper derives novel regret bounds.

---

[1] See Section 2 for a detailed description.

In our analysis, we define the notion of upper tail bounds, which measures the tightness of a family of distributions[2], and use it to bound the regret. This notion allows us to reduce the complexity of considering a family of delay distributions to considering only a single delay distribution. Then we use Bernstein inequality to control the effect of past actions on the observed reward of the current action that is being repeated. This allows us to obtain a regret upper bound in terms of the expected value of the upper tail bound. The use of upper tail bounds for studying regret in bandits with delayed feedback is novel and has not been considered in the literature earlier, to the best of our knowledge.

The main contributions of this paper can be summarized as follows

**1.** We introduce regret bounds for a stochastic CMAB problem with expected monotone and submodular rewards, a cardinality constraint, and composite anonymous feedback. Notably, this paper marks the first study of the regret bound any CMAB problem with composite delayed feedback, including CMAB with submodular rewards.

**2.** We investigate the ETCG algorithm from (Nie et al., 2022), detailing its performance in three feedback delay models: bounded adversarial delay, stochastic independent delay, and stochastic conditional independent delay. Specifically, this is the first study where the distribution of stochastic delay is permitted to vary over time. This introduces novel models for stochastic delayed composite anonymous feedback, which are more general than those previously explored in existing literature.

**3.** Our analysis reveals the cumulative $(1 - 1/e)$-regret of ETCG under specific bounds for each delay model. When comparing stochastic independent and conditional independent delays, the former showcases better regret bounds. Generalizing beyond specific parameters, our findings suggest a regret bound of $\tilde{O}(T^{2/3} + T^{1/3}\nu)$ across delay models.

**4.** Lastly, we showcase the adaptability of our analysis for delayed feedback in combinatorial bandits, given certain algorithmic conditions. Building on (Nie et al., 2022), we derive regret bounds for a meta-algorithm, highlighting its applicability to other CMAB problems such as submodular bandits with knapsack constraints (See (Nie et al., 2023)).

On the technical side, we define new generalized notions of delay and introduce the notion of upper tail bounds, which measures the tightness of a family of distributions. As discussed in Appendix A.2, algorithms designed for composite anonymous feedback, including those in our study, rely on the concept of repeating actions a sufficient number of times to minimize the impact of delay on the observed reward. We employ Bernstein's inequality to control the effect of previous actions on the observed reward of the current action that is being repeated. This approach enables us to establish an upper bound on regret, expressed in terms of the expected value of the upper tail bound.

Through simulations with synthetic data, we demonstrate that ETCG outperforms other full-bandit methods in the presence of delayed composite anonymous feedback.

## 2 Problem Statement

Let $T$ be the time horizon, $\Omega$ be the ground set of base arms, and $n := |\Omega|$. Also let $\mathcal{T}$ be a family of probability distributions on non-negative integers. At each time-step $t \geq 1$, the agent chooses an action $S_t$ from the set $\mathcal{S} = \{S | S \subseteq \Omega, \ |S| \leq k\}$, where $k$ is the a given positive integer.

The environment chooses a delay distribution $\delta_t \in \mathcal{T}$. The observation $x_t$ will be given by the formula

$$x_t = \sum_{i=1}^{t} f_i(S_i)\delta_i(t - i), \tag{1}$$

where $f_t(S)$ is sampled from $F_t(S)$, the stochastic reward function taking its values in $[0, 1]$. Moreover, we assume that $\mathbb{E}[F_t(S)] = f(S)$, where $f : 2^\Omega \to [0, 1]$ a monotone and submodular function. We will use $X_t$ to denote the random variable representing the observation at time $t$.

---

[2]See Assumption 1 and Lemma 1 for more details.

For $\alpha \in (0, 1]$, the $\alpha$-pseudo-regret is defined by

$$\mathcal{R}_\alpha := \sum_{t=1}^{T} \left( \alpha f(S^*) - f(S_t) \right),$$

where $S^* := \text{argmax}_{S \in \mathcal{S}} f(S)$ is the optimal action. Note that the choice of $\alpha = 1$ corresponds to the classical notion of pseudo-regret. When there is no ambiguity, we will simply refer to $\mathcal{R}_\alpha$ as the $\alpha$-regret or regret. In the offline problem with deterministic $f$, finding the optimal action $S^*$ is NP-hard. In fact, for $\alpha > 1 - 1/e$, (Feige, 1998) showed that finding an action which is at least as good as $\alpha f(S^*)$ is NP-hard. However, the standard greedy approach obtains a set which is at least as good as $(1 - 1/e)f(S^*)$ (Nemhauser et al., 1978). Therefore, throughout this paper, we will focus on minimizing $(1 - 1/e)$-regret and drop the subscript when there is no ambiguity.

We consider three settings: bounded adversarial delay and unbounded stochastic independent delay, and unbounded stochastic conditionally independent delay, described next.

**Example 1.** *To elaborate on the nature of the delay, let us ignore the combinatorial aspect of the problem for the moment and consider the following setting. A retailer, that sells both food and computer products, can buy an advertisement slot on an E-commerce platform, e.g., Amazon or eBay. This is a 2-armed bandit where we assume that the retailer buys an ad slot for a product at each time-step. We further assume that each time-step is a single day and the only information revealed to the retailer every day is the total added revenue as a result of the advertisements.*

*A delay distribution is a sequence of real numbers that add to one, e.g., $\delta = (0.9, 0.05, 0.05, 0, \cdots)$. Such a delay means that $90\%$ of the reward (increase in revenue as a result of the ads) is received immediately, while $5\%$ of the reward is received in each of the next 2 time-steps. Clearly it is not enough to consider a fixed delay distribution. Therefore we consider a situation where $\Delta$ is a random variable where $\delta$ is a realization of $\Delta$.*

*It is reasonable to assume that the effect of an ad for food is more immediately seen in the revenue compared to the effect of an ad for computer products. Therefore we may consider a setting where $\Delta_F$ is a random delay distribution corresponding to food and $\Delta_C$ correspond to computer products and $\Delta_F \neq \Delta_C$. This corresponds to the setting considered in (Wang et al., 2021) and (Garg & Akash, 2019).*

*Now assume that a sale for computer products, but not food, is going to start next week. Modeling this scenario means that $\Delta$ should change over time, but should also depend on the action, since only one of the actions is affected by the sales. This corresponds to Unbounded Stochastic Conditionally Independent Delay considered in our paper.*

*If we instead assume that the delay changes over time, but does not depend on the arm (for example if the retailer is selling different types of computer products), then this will be Unbounded Stochastic Independent Delay.*

*Finally, if delay is too complicated to be covered by previous settings, then we consider Bounded Adversarial Delay. For example, consider a scenario where different retailers pay the E-commerce platform for advertisement slots, but when the ad is shown depends on the buyers and the actions of other retailers, which can not be known in advance. The boundedness assumption guarantees that for each ad slot purchased, the effect on the revenue of the retailer will be limited to a fixed time, e.g. one month, from the purchase of the ad.*

### 2.1 Unbounded Stochastic Independent Delay

In the unbounded stochastic independent delay case, we assume that there is a sequence of random delay distributions $(\Delta_t)_{t=1}^{\infty}$ that is pair-wise independent, such that

$$X_t = \sum_{i=1}^{t} F_i(S_i) \Delta_i(t - i).$$

In other words, at each time-step $t$, the observed reward is based on all the actions that have been taken in the past and the action taken in time-step $i \leq t$ contributes to the observation proportional to the value of

the delay distribution at time $i$, $\Delta_i$, evaluated at $t - i$. We call this feedback model *composite anonymous unbounded stochastic independent delay feedback.*

To define $\Delta_t$, let $(\delta_i)_{i \in \mathcal{J}}$ be distributions chosen from $\mathcal{T}$, where $\mathcal{J}$ is a finite index set and each $\delta_i$ is represented by a vector of its probability mass function. Thus, $\delta_i(x) = \mathbb{P}(\delta_i = x)$, for all $x \geq 0$. Let $P_t$ be a random variables taking values in $\mathcal{J}$, where $P_t(i) = \mathbb{P}(P_t = i)$. Further, we define $\Delta_t(x) := \sum_{i \in \mathcal{J}} P_t(i)\delta_i(x)$, for all $x \geq 0$. Finally, $\Delta_t$ is defined as a vector $(\Delta_t(0), \Delta_t(1), \cdots)$. Note that $\sum_{i=0}^{\infty} \Delta_t(i) = 1$. The expectation of $\Delta_t$ over the randomness of $P_t$ is denoted by $\mathbb{E}_{\mathcal{T}}(\Delta_t)$ which is a distribution given $\delta_i$'s are distributions.

More generally, we may drop the assumption that $\mathcal{J}$ is finite and define $\Delta_t$ more directly as follows. Each $\Delta_t$ is a random variable taking values in the set $\mathcal{T}$. In other words, for all $x \geq 0$, the value of $\Delta_t(x) = \Delta_t(\{x\})$ is a random variable taking values in $[0, 1]$ such that $\sum_{i=0}^{\infty} \Delta_t(i) = \Delta_t(\{0, 1, 2, \cdots\}) = 1$. We define $\mathbb{E}_{\mathcal{T}}(\Delta_t)$ as the distribution over the set of non-negative integers for which we have

$$\forall x \geq 0, \quad \mathbb{E}_{\mathcal{T}}(\Delta_t)(\{x\}) = \mathbb{E}_{\mathcal{T}}(\Delta_t(\{x\})) \in [0, 1].$$

We will also explain these definitions by an example. Let $\mathcal{T}$ be a family of distributions supported on $\{0, 1, 2\}$. We choose $\mathcal{J} = \{1, 2\}$, with $\delta_i$ as the uniform distribution over $\{0, 1\}$ and $\delta_2$ as the uniform distribution over $\{0, 2\}$. Then, we have $\delta_1(0) = \delta_1(1) = 1/2$ and $\delta_2(0) = \delta_2(2) = 1/2$. Further, let $P_1$ be a random variable such that $P_1(1) + P_1(2) = 1$. Then, $\Delta_1(x) = \sum_{i=1,2} P_1(i)\delta_i(x)$ gives $\Delta_1(0) = P_1(1)/2 + P_1(2)/2$, $\Delta_1(1) = P_1(1)/2$ and $\Delta_1(2) = P_1(2)/2$.

Note that the independence implies that $\Delta_t$ can not depend on the action $S_t$, as this action depends on the history of observations, which is not independent from $(\Delta_j)_{j=1}^{t-1}$.

Without any restriction on the delay distributions, there may not be any reward within time $T$ and thus no structure of the rewards can be exploited. Thus, we need to have some guarantee that the delays do not escape to infinity. An appropriate formalization of this idea is achieved using the following tightness assumption.

**Assumption 1.** The family of distributions $(\mathbb{E}_{\mathcal{T}}(\Delta_t))_{t=1}^{\infty}$ is tight.

Recall that a family $(\delta_i)_{i \in I}$ is called tight if and only if for every positive real number $\epsilon$, there is an integer $j_\epsilon$ such that $\delta_i(\{x \geq j_\epsilon\}) \leq \epsilon$, for all $i \in I$. (See e.g. (Billingsley, 1995))

*Remark* 1. If $\mathcal{T}$ is tight, then $(\mathbb{E}_{\mathcal{T}}(\Delta_t))_{t=1}^{\infty}$ is trivially tight. Note that if $\mathcal{T}$ is finite, then it is tight. Similarly, if $(\mathbb{E}_{\mathcal{T}}(\Delta_t))_{t=1}^{\infty}$ is constant and therefore only takes one value, then it is tight. As a special case, if $(\Delta_t)_{t=1}^{\infty}$ is identically distributed, then $(\mathbb{E}_{\mathcal{T}}(\Delta_t))_{t=1}^{\infty}$ is constant and therefore tight.

To quantify the tightness of a family of probability distribution, we define the notion of *upper tail bound.*

**Definition 1.** Let $(\delta_i)_{i \in I}$ be a family of probability distributions over the set of non-negative integers. We say $\delta$ is an *upper tail bound* for this family if

$$\delta_i(\{x \geq j\}) \leq \delta(\{x \geq j\}),$$

for all $i \in I$ and $j \geq 0$.

In the following result (with proof in Appendix B), we show that the tightness and the existence of upper tail bounds are equivalent.

**Lemma 1.** *Let $(\delta_i)_{i \in I}$ be a family of probability distributions over the set of non-negative integers. Then this family is tight, if and only if it has an upper tail bound.*

A tail upper bound allows us to estimate and bound the effect of past actions on the current observed reward. More precisely, given an upper tail bound $\tau$ for the family $(\mathbb{E}_{\mathcal{T}}(\Delta_t))_{t=1}^{\infty}$, the effect of an action taken at time $i$ on the observer reward at $t$ is proportional to $\Delta_i(t - i)$, which can be bounded in expectation by $\tau$.

$$\mathbb{E}_{\mathcal{T}}(\Delta_i(t - i)) \leq \mathbb{E}_{\mathcal{T}}(\Delta_i(\{x \geq t - i\})) \leq \tau(\{x \geq t - i\}).$$

As we will see, only the expected value of the upper tail bound appears in the regret bound.

## 2.2 Unbounded Stochastic Conditionally Independent Delay

In the unbounded stochastic conditionally independent delay case, we assume that there is a family of random delay distributions $\{\Delta_{t,S}\}_{t \geq 1, S \in \mathcal{S}}$ such that for any $S \in \mathcal{S}$, the sequence $(\Delta_{t,S})_{t=1}^{\infty}$ is pair-wise independent and

$$X_t = \sum_{i=1}^{t} F_i(S_i)\Delta_{i,S_i}(t - i).$$

We call this feedback model *composite anonymous unbounded stochastic conditionally independent delay feedback*.

In this case the delay $\Delta_t = \Delta_{t,S_t}$ can depend on the action $S_t$, but conditioned on the current action, it is independent of (some of the) other conditional delays. Similar to the stochastic independent delay setting, we assume that the sequence $\{\mathbb{E}_{\mathcal{T}}(\Delta_{t,S})\}_{t \geq 1, S \in \mathcal{S}}$ is tight.

*Remark* 2. In previously considered stochastic composite anonymous feedback models (e.g., (Wang et al., 2021; Garg & Akash, 2019)), the delay distribution is independent of time. In other words, every action $S$ has a corresponding random delay distribution $\Delta_S$, and the sequence $(\Delta_{t,S})_{t=1}^{\infty}$ is independent and identically distributed. Therefore, the number of distributions in the set $\{\mathbb{E}_{\mathcal{T}}(\Delta_{t,S})\}_{t \geq 1, S \in \mathcal{S}}$ is less than or equal to the number of arms, which is finite. Hence the family $\{\mathbb{E}_{\mathcal{T}}(\Delta_{t,S})\}_{t \geq 1, S \in \mathcal{S}}$ is tight.

## 2.3 Bounded Adversarial Delay

In the bounded adversarial delay case, we assume that there is an integer $d \geq 0$ such that for all $\delta \in \mathcal{T}$, we have $\delta(\{x > d\}) = 0$. Here we have

$$X_t = \sum_{i=\max\{1, t-d\}}^{t} F_i(S_i)\delta_i(t - i),$$

where $(\delta_t)_{t=1}^{\infty}$ is a sequence of distributions in $\mathcal{T}$ chosen by the environment. Here we used $\delta$ instead of $\Delta$ to emphasize the fact that these distributions are not chosen according to some random variable with desirable properties. In fact, the environment may choose $\delta_t$ non-obliviously, that is with the full knowledge of the history up to the time-step $t$. We call this feedback model *composite anonymous bounded adversarial delay feedback*.

---

**Algorithm 1** ETCG algorithm

---

**Input:** Set of base arms $\Omega$, horizon $T$, cardinality constraint $k$
**Assumption:** $n \leq T$
1: $S^{(0)} \leftarrow \emptyset$, $n \leftarrow |\Omega|$
2: $m \leftarrow \lceil (T/n)^{2/3} \rceil$
3: **for** phase $i \in \{1, 2, \cdots, k\}$ **do**
4:     **for** arm $a \in \Omega \setminus S^{(i-1)}$ **do**
5:         Play $S^{(i-1)} \cup \{a\}$ arm $m$ times
6:         Calculate the empirical mean $\bar{x}_{i,a}$
7:     **end for**
8:     $a_i \leftarrow \text{argmax}_{a \in \Omega \setminus S^{(i-1)}} \bar{x}_{i,a}$
9:     $S^{(i)} \leftarrow S^{(i-1)} \cup \{a_i\}$
10: **end for**
11: **for** remaining time **do**
12:     Play action $S^{(k)}$
13: **end for**

---

## 3 Regret Analysis with Delayed Feedback

For analyzing the impact of delay, we use the algorithm Explore-Then-Commit-Greedy (ETCG) algorithm, as proposed in (Nie et al., 2022). We start with $S^{(0)} = \emptyset$ in phase $i = 0$. In each phase $i \in \{1, \cdots, k\}$, we go over the list of all base arms $\Omega \setminus S^{(i-1)}$. For each such base arm, we take the action $S^{(i-1)} \cup \{a\}$ for $m = \lceil (T/n)^{2/3} \rceil$ times and store the empirical mean in the variable $\bar{X}_{i,a}$. Afterwards, we let $a_i$ to be the base arm which corresponded to the highest empirical mean and define $S^{(i)} := S^{(i-1)} \cup \{a_i\}$. After the end of phase $k$, we keep taking the action $S^{(k)}$ for the remaining time. The algorithm is summarized in Algorithm 1.

We now provide the main results of the paper that shows the regret bound of Algorithm 1 with delayed composite feedback for different feedback models. We define two main events that control the delay and the randomness of the observation. Let $I = \{(i, a) \mid 1 \leq i \leq k, a \in \Omega \setminus S^{i-1}\}$, and define

$$\mathcal{E} := \left\{|\bar{F}_{i,a} - f(S^{i-1} \cup \{a\})| \leq \text{rad} \mid (i, a) \in I\right\}, \text{ and } \mathcal{E}_d' := \left\{|\bar{F}_{i,a} - \bar{X}_{i,a}| \leq \frac{2d}{m} \mid (i, a) \in I\right\},$$

where rad, $d > 0$ are real numbers that will be specified later. We may drop the subscript $d$ when it is clear from the context. When $\mathcal{E}$ happens, the average observed reward associated with each arm stays close its expectation, which is the value of the submodular function. When $\mathcal{E}'_d$ happens, the average observed reward for each arm remains close to the average total reward associated with playing that arm. The next result bounds the regret as:

**Theorem 1.** *For all $d > 0$, we have*

$$\mathbb{E}(\mathcal{R}) \leq mnk + 2kT\,\mathrm{rad} + \frac{4kTd}{m} + 2nkT\exp(-2m\,\mathrm{rad}^2) + T(1 - \mathbb{P}(\mathcal{E}'_d)).$$

See Appendix C for a detailed proof. To obtain the regret bounds for different settings, we need to find lower bounds for $\mathbb{P}(\mathcal{E}'_d)$ and use Theorem 1.

**Theorem 2** (Bounded Adversarial Delay)**.** *If the delay is uniformly bounded by $d$, then we have*

$$\mathbb{E}(\mathcal{R}) = O(kn^{1/3}T^{2/3}(\log(T))^{1/2}) + O(kn^{2/3}T^{1/3}d).$$

*Proof.* The detailed proof is provided in Appendix D. Here, we describe the proof outline. In this setting, there is an integer $d \geq 0$ such that $\delta_t(\{x > d\}) = 0$, for all $t \geq 1$. Therefore, for any $m$ consecutive time-steps $t_{i,a} \leq t \leq t'_{i,a}$, the effect of delay may only be observed in the first $d$ and the last $d$ time-steps. It follows that $\left| \sum_{t=t_{i,a}}^{t'_{i,a}} X_t - \sum_{t=t_{i,a}}^{t'_{i,a}} F_t \right| \leq 2d$, for all $(i,a) \in I$. Therefore, in this case, we have $\mathbb{P}(\mathcal{E}'_d) = 1$. Note that we are not making any assumptions about the delay distributions. Therefore, the delay may be chosen by an adversary with the full knowledge of the environment, the algorithm used by the agent and the history of actions and rewards. Plugging this in the bound provided by Theorem 1 completes the proof. $\square$

We note that $\widetilde{O}(T^{2/3})$ is the best known bound for the problem in the absence of the delayed feedback, and the result here demonstrate an additive impact of the delay on the regret bounds.

**Theorem 3** (Stochastic Independent Delay)**.** *If the delay sequence is stochastic and independent and tight in expectation, then we have*

$$\mathbb{E}(\mathcal{R}) = O(kn^{1/3}T^{2/3}\log(T)) + O(kn^{2/3}T^{1/3}\mathbb{E}(\tau)),$$

*where $\tau$ is an upper tail bound for $\{\mathbb{E}_{\mathcal{T}}(\Delta_t)\}_{t=1}^{\infty}$.*

*Proof.* The detailed proof is provided in Appendix E. Here, we describe the proof outline. We start by defining the random variables $C_{i,a} = \sum_{j=1}^{t'_{i,a}} \Delta_j(\{x > t'_{i,a} - j\})$, for all $(i,a) \in I$. This random variable measure the effect of actions taken up to $t'_{n,i}$ on the observed rewards after $t'_{n,i}$. In fact, we will see that $m|\bar{X}_{i,a} - \bar{F}_{i,a}|$ may be bounded by the sum of two terms. One $C_{i,a}$ which bounds the amount of reward that "escapes" from the time interval $[t_{i,a}, t'_{i,a}]$. The second one $C_{i',a'}$, where $(i',a')$ corresponds to the action taken before $S^{i-1} \cup \{a\}$. This bound corresponds to the total of reward of the past actions that is observed during $[t_{i,a}, t'_{i,a}]$. Therefore, in order for the event $\mathcal{E}'_d$ to happen, it is sufficient to have $C_{i,a} \leq d$, for all $(i,a) \in I$. Since $C_{i,a}$ is a sum of independent random variables, we may use Bernstein's inequality to see that $\mathbb{P}(C_{i,a} > \mathbb{E}(C_{i,a}) + \lambda) \leq \exp\left(-\frac{\lambda^2}{2(\mathbb{E}(\tau) + \lambda/3)}\right)$. It follows from the definition that $\mathbb{E}(C_{i,a}) \leq \mathbb{E}(\tau)$. Therefore, by setting $d = \mathbb{E}(\tau) + \lambda$, and performing union bound on the complement of $\mathcal{E}'_d$ gives $\mathbb{P}(\mathcal{E}'_d) \geq 1 - nk \exp\left(-\frac{\lambda^2}{2(\mathbb{E}(\tau) + \lambda/3)}\right)$. Plugging this in Theorem 1 and choosing appropriate $\lambda$ gives us the desired result. $\square$

**Theorem 4** (Stochastic Conditionally Independent Delay)**.** *If the delay sequence is stochastic, conditionally independent and tight in expectation, then we have*

$$\mathbb{E}(\mathcal{R}) = O(k^2 n^{4/3} T^{2/3} \log(T)) + O(k^2 n^{5/3} T^{1/3} \mathbb{E}(\tau))$$

*where $\tau$ is an upper tail bound for $(\mathbb{E}_{\mathcal{T}}(\Delta_{t,S}))_{t \geq 1, S \in \mathcal{S}}$.*

*Proof.* The detailed proof is provided in Appendix E and is similar to the proof of Theorem 3. The main difference is that here we define $C'_{i,a} = \sum_{j=t_{i,a}}^{t'_{i,a}} \Delta_j(\{x > t'_{i,a} - j\})$, instead of $C_{i,a}$. Note that the sum here is only over the time-steps where the action $S^{i-1} \cup \{a\}$ is taken. Therefore $C'_{i,a}$ is the sum of $m$ independent term. On the other hand, when we try to bound $m|\bar{X}_{i,a} - \bar{F}_{i,a}|$, we decompose it into the amount of total reward that "escapes" from the time interval $[t_{i,a}, t'_{i,a}]$ and the contribution of all the time intervals of the form $[t_{i',a'}, t'_{i',a'}]$ in the past. Since the total number of such intervals is bounded by $nk$, here we find the probability that $C'_{i,a} \leq \frac{2d}{nk}$ instead of $C_{i,a} \leq d$ as we did in the proof of Theorem 3. This is the source of the multiplicative factor of $nk$ which appears behind the regret bound of this setting when compared to the stochastic independent delay setting. $\square$

## 4 Beyond Monotone Submodular Bandits

We note that (Nie et al., 2023) provided a generalized framework for combinatorial bandits with full bandit feedback, where under a robustness guarantee, explore-then-commit (ETC) based algorithm have been used to get provable regret guarantees. More precisely, let $\mathcal{A}$ be an algorithm for the combinatorial optimization problem of maximizing a function $f : \mathcal{S} \to \mathbb{R}$ over a finite domain $\mathcal{S} \subseteq 2^\Omega$ with the knowledge that $f$ belongs to a known class of functions $\mathcal{F}$. for any function $\hat{f} : \mathcal{S} \to \mathbb{R}$, let $\mathcal{S}_{\mathcal{A},\hat{f}}$ denote the output of $\mathcal{A}$ when it is run with $\hat{f}$ as its value oracle. The algorithm $\mathcal{A}$ called $(\alpha, \delta)$-robust if for any $\epsilon > 0$ and any function $\hat{f}$ such that $|f(S) - \hat{f}(S)| < \epsilon$ for all $S \in \mathcal{S}$, we have

$$f(S_{\mathcal{A},\hat{f}}) \geq \alpha f(S^*) - \delta\epsilon.$$

It is shown in (Nie et al., 2023) that if $\mathcal{A}$ is $(\alpha, \delta)$-robust, then the C-ETC algorithm achieves $\alpha$-regret bound of $O(N^{1/3}\delta^{2/3}T^{2/3}(\log(T))^{1/2})$, where $N$ is an upper-bound for the number of times $\mathcal{A}$ queries the value oracle (the detailed result and algorithm is given in Appendix G). In this work, we show that the result could be extended directly with delayed composite anonymous bandit feedback. The proof requires small changes, and are detailed in Appendix G. If $\mathcal{A}$ is $(\alpha, \delta)$-robust, then the results with bandit feedback are as follows.

**Theorem 5.** *If the delay is uniformly bounded by $d$, then we have*

$$\mathbb{E}(\mathcal{R}_\alpha) = O(N^{1/3}\delta^{2/3}T^{2/3}(\log(T))^{1/2}) + O(N^{2/3}\delta^{1/3}T^{1/3}d).$$

**Theorem 6.** *If the delay sequence is stochastic, then we have*

$$\mathbb{E}(\mathcal{R}_\alpha) = O(N^{1/3}\delta^{2/3}T^{2/3}\log(T)) + O(N^{2/3}\delta^{1/3}T^{1/3}\mathbb{E}(\tau)),$$

*where $\tau$ is an upper tail bound for $(\mathbb{E}_\mathcal{T}(\Delta_t))_{t=1}^\infty$.*

**Theorem 7.** *If the delay sequence is stochastic and conditionally independent, then we have*

$$\mathbb{E}(\mathcal{R}_\alpha) = O(N^{4/3}\delta^{2/3}T^{2/3}\log(T)) + O(N^{5/3}\delta^{1/3}T^{1/3}\mathbb{E}(\tau)),$$

*where $\tau$ is an upper tail bound for $(\mathbb{E}_\mathcal{T}(\Delta_t))_{t=1}^\infty$.*

This shows that the proposed approach in this paper that deals with feedback could be applied on wide variety of problems. The problems that satisfy the robustness guarantee include submodular bandits with knapsack constraints and submodular bandits with cardinality constraints (considered earlier).

## 5 Experiments

In our experiments, we consider two classes of submodular functions (Linear (F1) and Weight Cover (F2)) and and six types of delay (No Delay (D1), two setups of Stochastic Independent Delay (D2, D3), two setups of Stochastic Conditionally Independent Delay (D4, D5), and Adversarial Delay (D6)). For linear function (F1), we choose $F(S) := \frac{1}{k} \sum_{a \in S} g(a) + N^c(0, 0.1)$ where $N^c(0, 0.1)$ is the truncated normal distribution with mean

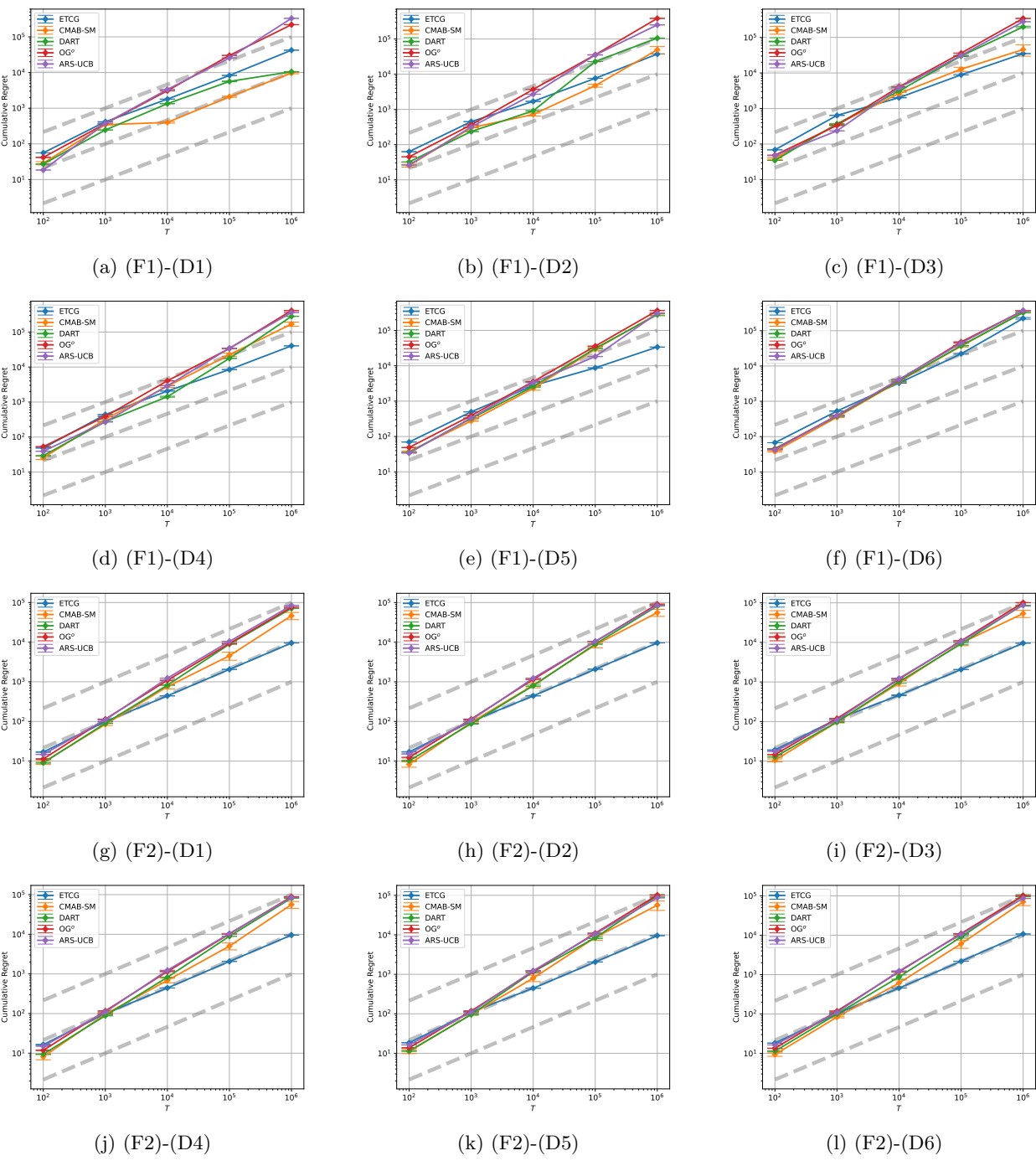

Figure 1: This plot shows the average cumulative 1-regret over horizon for each setting in the log-log scale. The dashed lines are $y = aT^{2/3}$ for $a \in \{0.1, 1, 10\}$. Note that (F1) is a linear function and (D1) is the setting with no delay. Moreover, (D2) corresponds to a delay setting where delay distributions are concentrated near zero and decay exponentially.

0 and standard deviation 0.1, truncated to the interval $[-0.1, 1.0]$, $n = 20$ and $k = 4$ and choose $g(a)$ uniformly from $[0.1, 0.9]$, for all $a \in \Omega$. For weight cover function (F2), we choose $f_t(S) := \frac{1}{k} \sum_{j \in J} w_t(j) \mathbb{1}_{S \cap C_j \neq \emptyset}$, where $w_t(j) = U([0, j/5])$ be samples uniformly from $[0, j/5]$ for $j \in 1, 2, 3, 4$, $n = 20$ and $k = 4$, $(C_j)_{j \in J}$ is a partition of $\Omega$ where $\Omega$ is divided into 4 categories of sizes $6, 6, 6, 2$. Stochastic set cover may be viewed as a simple model for product recommendation, and more details on the function choices is in Appendix H. For

the delay types, (D2) assumes $\Delta_t(i) = (1 - X_t)X_t^i$, where $(X_t)_{t=1}^{\infty}$ is an i.i.d sequence of random variables with the uniform distribution $U([0.5, 0.9])$ for all $t \geq 1$ and $i \geq 0$. (D3) assumes $\Delta_t$ is a distribution over $[10, 30]$ is sampled uniformly from the probability simplex using the flat Dirichlet distribution for all $t \geq 0$. (D4) assumes $\Delta_t(i) = (1 - Y_t)Y_t^i$, where $Y_t = 0.5 + f_t * 0.4 \in [0.5, 0.9]$ for all $t \geq 1$ and $i \geq 0$. (D5) assumes $\Delta_t$ as deterministic taking value at $l_t = \lfloor 20f_t \rfloor + 10$ for all $t$. (D6) ssumes $\Delta_t$ as deterministic taking value at $l_t = \lfloor 20x_{t-1} \rfloor + 10$ with $l_1 = 15$ for all $t > 1$. The setups are detailed in Appendix H.

For comparisons, we use the baselines of **CMAB-SM** (Agarwal et al., 2022), **DART** (Agarwal et al., 2021), **OG**$^o$ (Streeter & Golovin, 2008), and **ARS-UCB** (Wang et al., 2021), with details in Appendix H. We use $n = 20$ base arms and cardinality constraint $k = 4$. We run each experiment for different time horizons $T = \{10^2, 10^3, 10^4, 10^5, 10^6\}$. For each horizon, we run the experiment 10 times. In these experiments, ETCG outperforms all other baselines for the weighted cover function by almost an order of magnitude. The linear submodular function satisfies the conditions under which DART and CMAB-SM were designed. However, the weighed cover function does not satisfy such conditions and therefore more difficult for those algorithms to run. In both cases, we see that any kind of delay worsens the performance of DART and CMAB-SM compared to ETCG. OG$^o$ explores actions (including those with cardinality smaller then $k$) with a constant probability, which could account for its lower performance compared to ETCG, DART, and CMAB-SM.

While ARS-UCB does not perform well in these experiments, it should be noted that, given enough time, it should outperform ETCG. Also note that ARS-UCB has a linear storage complexity with respect to its number of arms. This translates to an $O(\binom{n}{k})$ storage complexity in the combinatorial setting. Therefore, even for $n = 50$ and $k = 25$, it would require hundreds of terabytes of storage to run. In these experiments, we have $n = 20$ and $k = 4$, so it has only $\binom{20}{4} = 4845$ arms.

## 6    Conclusion

This paper considered the problem of combinatorial multiarmed bandits with stochastic submodular (in expectation) rewards and delayed composite anonymous bandit feedback and provides first regret bound results for this setup. Three models of delayed feedback: bounded adversarial, stochastic independent, and stochastic conditionally independent are studied, and regret bounds are derived for each of the delay models. The regret bounds demonstrate an additive impact of delay in the regret term.

**Limitations:**    This paper demonstrates an additive impact of delay in the regret term, where the non-delay term is the state-of-the-art regret bound. We note that this state-of-the-art regret bound is $\tilde{O}(T^{2/3})$, while there is no matching lower bound. Further, our result shows $\tilde{O}(T^{1/3})$ dependence in the additive delay term, while exploring optimality of such dependence is open.

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

# A   Other Related Works

We note that this is the first work to derive regret bounds for CMAB with submodular and monotone rewards and delayed feedback. Thus, the most related work can be divided into the results for CMAB with submodular and monotone rewards, and that for MAB with delayed feedback, as will be described next.

## A.1   Combinatorial Submodular Bandits

CMABs have been widely studied due to multiple applications. While the problem of CMAB is general and there are multiple studies that do not use submodular rewards (Agarwal et al., 2021; 2022; Dani et al., 2008; Rejwan & Mansour, 2020), we consider CMAB with monotone and submodular rewards. The assumption of monotonicity and submodularity in reward functions is common in the literature (Streeter et al., 2009; Niazadeh et al., 2021; Nie et al., 2022). For CMAB with monotone and submodular rewards, without any further constraints, the optimal selection will be the entire set. Thus, additional assumptions are introduced in the model, including cardinality constraint (Nemhauser et al., 1978) and knapsack constraints (Sviridenko, 2004). This paper considers CMAB with submodular and monotone rewards and cardinality constraint.

Further, we note that feedback pays an important role in CMAB decision making. CMAB with submodular and monotone rewards and cardinality constraint has been studied with semi-bandit feedback (Lin et al., 2015; Niazadeh et al., 2021; Zhang et al., 2019; Zhu et al., 2021; Chen et al., 2018a; Takemori et al., 2020). The semi-bandit feedback setting provides more information as compared to the full-bandit setting. The same is true for contextual bandit feedback (Yue & Guestrin, 2011; Chen et al., 2018b) as well. Here we consider the full-bandit (or bandit) feedback without any additional feedback. CMAB with submodular and monotone rewards, cardinality constraint, and full-bandit feedback has been studied in both adversarial setting (Niazadeh et al., 2021) and in stochastic setting (Nie et al., 2022). This paper studies the stochastic setting.

It is worth noting that for submodular bandits, the stochastic reward case is not a special case of the adversarial reward case and the guarantees for the stochastic reward case are not necessarily better than the adversarial reward case. In the adversarial setting, the environment chooses a sequence of monotone and submodular functions $\{f_1, \cdots, f_T\}$. This is incompatible with the stochastic reward setting since we only require the set function $f_t$ to be monotone and submodular in expectation. Thus, the results on adversarial submodular bandits will not lead to results for the stochastic submodular setting.

These works for CMAB do not study regret bound with delayed feedback, which is the focus of this paper.

## A.2   Bandits With Delayed Rewards

The bandit problem with (non-anonymous) delayed feedback has been studied extensively (Mesterharm, 2005; Agarwal & Duchi, 2011; Desautels et al., 2014; Dudik et al., 2011; Joulani et al., 2013). In the non-anonymous setting, the reward will be delayed and at each time-step, the agent observers a set of the form $\{(t, r_t) \mid t \in I_t\}$ where $I_t$ is a set of time-steps in the past. In the aggregated anonymous setting, first studied by (Pike-Burke et al., 2018), the reward for each arm is obtained at some point in the future, so that the agent will receive the aggregated reward for some of the past actions at each time-step. (Cesa-Bianchi et al., 2018) extended the reward model so that the reward of an action is not immediately observed by the agent, but rather spread over at most $d$ consecutive steps in an adversarial way. However, they also assumed that the bandit is adversarial. (Garg & Akash, 2019) considered the stochastic case and provided an algorithm with a sub-linear regret bound of $\tilde{O}(n^{1/2}T^{1/2}) + O(n \log(T)d)$. In this setting, for each arm $a$, the there is a random distribution $\Delta_a$ over the set $\{0, 1, \cdots, d\}$ and at each time-step, when the agent plays $a$, the delay is sampled from $\Delta_a$. (Wang et al., 2021) also considers unbounded delay and proves the regret bound of $\tilde{O}(n^{1/2}T^{1/2}) + O(\nu)$, where $\nu$ depends on the delay distribution and $n$ but not on $T$. They also considered the case with adversarial but bounded delay and proved a regret bound of $\tilde{O}(n^{1/2}T^{2/3}) + O(T^{2/3}d)$. We note in all of the works addressing composite anonymous feedback, including ours, a key idea is to repeat actions enough times so that we can extract meaningful information. This is not always necessary in other types of delay. In particular, if delay is not anonymous, there is no need to repeat actions since we will eventually know the reward for each action. Except for ETCG of (Nie et al., 2022) and more generally, instances of

the meta-algorithm C-ETC algorithm of (Nie et al., 2023), other algorithms discussed here for combinatoral bandits do not repeat actions and therefore there is little hope of them achieving desirable results in the presence of composite anonymous delay.

Note that, in the submodular setting, any algorithm that does not exploit the combinatorial structure of the arms must take at least every action once which can be suboptimal since the number of arms is at least $\binom{n}{k}$ which grows exponentially.

In this paper, we extend the delay model further by letting the random delay distribution also depend on time (See Example 1 and Remark 2 for more details).

## B   Proof of Lemma 1

*Proof.* If an upper tail bound $\delta$ exists, then we may simply define

$$j_\epsilon := \min\{j \mid \delta(\{x \geq j\}) \leq \epsilon\},$$

to see that the family is tight. Next we assume that the family is tight and prove the existence of an upper tail bound.

Let $\delta$ be the measure defined by

$$\forall j \geq 0, \quad \delta(j) := \sup_{i \in I} \delta_i(\{x \geq j\}) - \sup_{i \in I} \delta_i(\{x \geq j+1\}).$$

Clearly we have $\delta(j) \geq 0$, for all $j \geq 0$. To show that $\delta$ is a probability distribution, we sum the terms and see that

$$\delta(\{t \mid a \leq t \leq b\}) = \sum_{t=a}^{b} \delta(t) = \sum_{t=a}^{b} \left( \sup_{i \in I} \delta_i(\{x \geq t\}) - \sup_{i \in I} \delta_i(\{x \geq t+1\}) \right)$$
$$= \sup_{i \in I} \delta_i(\{x \geq a\}) - \sup_{i \in I} \delta_i(\{x \geq b+1\})$$

According to the definition of tightness, for all $\epsilon > 0$ and $b \geq j_\epsilon$, we have

$$\delta(\{t \mid a \leq t \leq b\}) = \sup_{i \in I} \delta_i(\{x \geq a\}) - \sup_{i \in I} \delta_i(\{x \geq b+1\}) \geq \sup_{i \in I} \delta_i(\{x \geq a\}) - \epsilon.$$

Hence we have

$$\delta(\{t \geq 0\}) = \lim_{j \to \infty} \delta(\{t \mid 0 \leq t \leq j\}) = 1 - \lim_{j \to \infty} \sup_{i \in I} \delta_i(\{x \geq j+1\}) = 1.$$

Finally, to see that $\delta$ is indeed an upper tail bound, we note that

$$\delta(\{t \mid a \leq t \leq b\}) = \sup_{i \in I} \delta_i(\{x \geq a\}) - \sup_{i \in I} \delta_i(\{x \geq b+1\}) \leq \sup_{i \in I} \delta_i(\{x \geq a\}).$$

Therefore

$$\delta(\{t \geq a\}) = \lim_{b \to \infty} \delta(\{t \mid a \leq t \leq b\}) \leq \sup_{i \in I} \delta_i(\{x \geq a\}). \qquad \square$$

## C   Lemmas used in the proofs

In this section, we will provide the Lemmas that will be used in the proof of the main results. Let $t_{i,a}$ denote the first time-step where the action $S^{(i-1)} \cup \{a\}$ is played in the exploration phase and let $t'_{i,a} := t_{i,a} + m - 1$ be the last such time-step. Therefore, we have

$$\bar{F}_{i,a} = \frac{1}{m} \sum_{t=t_{i,a}}^{t'_{i,a}} F_t, \quad \bar{X}_{i,a} = \frac{1}{m} \sum_{t=t_{i,a}}^{t'_{i,a}} X_t.$$

Similarly, the realized value of these random variables are

$$\bar{f}_{i,a} = \frac{1}{m} \sum_{t=t_{i,a}}^{t'_{i,a}} f_t, \quad \bar{x}_{i,a} = \frac{1}{m} \sum_{t=t_{i,a}}^{t'_{i,a}} x_t.$$

For any phase $i$ and arm $a \in \Omega \setminus S^{i-1}$, define the event

$$\mathcal{E}_{i,a} := \left\{ |\bar{F}_{i,a} - f(S^{i-1} \cup \{a\})| \le \mathrm{rad} \right\},$$

where rad is a non-negative real number to be specified later. Using these events, we define

$$\mathcal{E}_i := \bigcap_{a \in \Omega \setminus S^{(i-1)}} \mathcal{E}_{i,a}, \quad \mathcal{E} := \bigcap_{i=1}^{k} \mathcal{E}_i.$$

**Lemma 2.** *We have*

$$\mathbb{P}(\mathcal{E}) \ge 1 - 2nk \exp(-2m \, \mathrm{rad}^2).$$

*Proof.* We have $F_t \in [0,1]$. Therefore, using Hoeffding's inequality, we have

$$\mathbb{P}(|\bar{F}_{i,a} - f(S^{i-1} \cup \{a\})| > \mathrm{rad}) = \mathbb{P}\left( \left| \sum_{t_{i,a}}^{t'_{i,a}} F_t - mf(S^{i-1} \cup \{a\}) \right| > m \, \mathrm{rad} \right)$$

$$\le 2 \exp\left( -\frac{2(m \, \mathrm{rad})^2}{m} \right) = 2 \exp(-2m \, \mathrm{rad}^2).$$

Hence

$$\begin{aligned}
\mathbb{P}(\mathcal{E}) &= \mathbb{P}\left( \bigcap_{i,a}^{k} \mathcal{E}_{i,a} \right) \\
&= 1 - \mathbb{P}\left( \bigcup_{i,a}^{k} (\mathcal{E}_{i,a})^c \right) \\
&\ge 1 - \sum_{i,a} \mathbb{P}((\mathcal{E}_{i,a})^c) \\
&= 1 - \sum_{i,a} \mathbb{P}(|\bar{F}_{i,a} - f(S^{i-1} \cup \{a\})| > \mathrm{rad}) \\
&\ge 1 - \sum_{i,a} 2 \exp(-2m \, \mathrm{rad}^2) \\
&\ge 1 - 2nk \exp(-2m \, \mathrm{rad}^2).
\end{aligned}$$
□

Next we define another set of events where the delay is controlled. Let

$$\mathcal{E}'_{d,i,a} := \left\{ |\bar{X}_{i,a} - \bar{F}_{i,a}| \le \frac{2d}{m} \right\},$$

for some $d > 0$ which will be specified later. Similar to above, we use these events to define

$$\mathcal{E}'_{d,i} := \bigcap_{a \in \Omega \setminus S^{(i-1)}} \mathcal{E}_{d,i,a}, \quad \mathcal{E}'_d := \bigcap_{i=1}^{k} \mathcal{E}'_{d,i}.$$

Note that $\mathcal{E}'_d$ can happen even if the delay is not bounded. Later in Lemmas 5 and 6, we will find lower bounds on the probability of $\mathcal{E}'_d$ in both the adversarial and the stochastic setting.

**Lemma 3.** *Under the event $\mathcal{E} \cap \mathcal{E}'_d$, for all $1 \leq i \leq k$ and $d > 0$, we have*

$$f(S^{(i)}) - f(S^{(i-1)}) \geq \frac{1}{k}\left[f(S^*) - f(S^{(i-1)})\right] - 2\,\mathrm{rad} - \frac{4d}{m}.$$

*Proof.* Recall that $a_i$ is the sole element in $S^i \setminus S^{i-1}$. That is,

$$a_i = \mathrm{argmax}_{a \in \Omega \setminus S^{(i-1)}}\, \bar{x}_{i,a}.$$

Define

$$a_i^* := \mathrm{argmax}_{a \in \Omega \setminus S^{(i-1)}}\, f(S^{i-1} \cup \{a\}).$$

Then we have

$$
\begin{aligned}
f(S^i) = f(S^{i-1} \cup \{a_i\}) \\
\geq \bar{f}_{i,a_i} - \mathrm{rad} && \text{(definition of } \mathcal{E}) \\
\geq \bar{x}_{i,a_i} - \frac{2d}{m} - \mathrm{rad} && \text{(definition of } \mathcal{E}') \\
\geq \bar{x}_{i,a_i^*} - \frac{2d}{m} - \mathrm{rad} && \text{(definition of } a_i^*) \\
\geq \bar{f}_{i,a_i^*} - \frac{4d}{m} - \mathrm{rad} && \text{(definition of } \mathcal{E}') \\
\geq f(S^{i-1} \cup \{a_i^*\}) - \frac{4d}{m} - 2\,\mathrm{rad}. && \text{(definition of } \mathcal{E})
\end{aligned}
$$

Hence we have

$$f(S^i) - f(S^{i-1}) \geq f(S^{i-1} \cup \{a_i^*\}) - f(S^{i-1}) - \frac{4d}{m} - 2\,\mathrm{rad}.$$

Therefore

$$
\begin{aligned}
f(S^i) - f(S^{i-1}) &\geq f(S^{i-1} \cup \{a_i^*\}) - f(S^{i-1}) - \frac{4d}{m} - 2\,\mathrm{rad} \\
&= \max_{a \in \Omega \setminus S^{(i-1)}} f(S^{i-1} \cup \{a\}) - f(S^{i-1}) - \frac{4d}{m} - 2\,\mathrm{rad} && \text{(definition of } a_i^*) \\
&\geq \max_{a \in S^* \setminus S^{(i-1)}} f(S^{i-1} \cup \{a\}) - f(S^{i-1}) - \frac{4d}{m} - 2\,\mathrm{rad} && (S^* \subseteq \Omega) \\
&\geq \frac{1}{|S^* \setminus S^{i-1}|} \sum_{a \in S^* \setminus S^{(i-1)}} f(S^{i-1} \cup \{a\}) - f(S^{i-1}) - \frac{4d}{m} - 2\,\mathrm{rad} && \text{(maximum} \geq \text{mean)} \\
&= \frac{1}{|S^* \setminus S^{i-1}|} \sum_{a \in S^* \setminus S^{(i-1)}} \left[f(S^{i-1} \cup \{a\}) - f(S^{i-1})\right] - \frac{4d}{m} - 2\,\mathrm{rad} \\
&\geq \frac{1}{k} \sum_{a \in S^* \setminus S^{(i-1)}} \left[f(S^{i-1} \cup \{a\}) - f(S^{i-1})\right] - \frac{4d}{m} - 2\,\mathrm{rad} && (|S^* \setminus S^{i-1}| \leq |S^*| = k) \\
&\geq \frac{1}{k}\left[f(S^*) - f(S^{i-1})\right] - \frac{4d}{m} - 2\,\mathrm{rad},
\end{aligned}
$$

where the last line follows from a well-known inequality for submodular functions. $\square$

**Corollary 1.** *Under the event $\mathcal{E} \cap \mathcal{E}'_d$, for all $d > 0$, we have*

$$f(S^{(k)}) \geq (1 - \frac{1}{e})f(S^*) - 2k\,\mathrm{rad} - \frac{4kd}{m}.$$

*Proof.* Using Lemma 3, we have

$$f(S^i) \geq f(S^{i-1}) + \frac{1}{k}(f(S^*) - f(S^{i-1})) - 2\,\mathrm{rad} - \frac{4d}{m} = \left[\frac{1}{k}f(S^*) - 2\,\mathrm{rad} - \frac{4d}{m}\right] + (1 - \frac{1}{k})f(S^{i-1}).$$

Applying this inequality recursively, we get

$$
\begin{aligned}
f(S^k) &\geq \left[\frac{1}{k}f(S^*) - 2\,\mathrm{rad} - \frac{4d}{m}\right] + (1 - \frac{1}{k})f(S^{k-1}) \\
&\geq \left[\frac{1}{k}f(S^*) - 2\,\mathrm{rad} - \frac{4d}{m}\right] + (1 - \frac{1}{k})\left(\left[\frac{1}{k}f(S^*) - 2\,\mathrm{rad} - \frac{4d}{m}\right] + (1 - \frac{1}{k})f(S^{k-2})\right) \\
&= \left[\frac{1}{k}f(S^*) - 2\,\mathrm{rad} - \frac{4d}{m}\right]\sum_{l=0}^{1}(1 - \frac{1}{k})^l + (1 - \frac{1}{k})^2 f(S^{k-2}) \\
&\vdots \\
&\geq \left[\frac{1}{k}f(S^*) - 2\,\mathrm{rad} - \frac{4d}{m}\right]\sum_{l=0}^{k-1}(1 - \frac{1}{k})^l + (1 - \frac{1}{k})^k f(S^0) \\
&= \left[\frac{1}{k}f(S^*) - 2\,\mathrm{rad} - \frac{4d}{m}\right]\sum_{l=0}^{k-1}(1 - \frac{1}{k})^l
\end{aligned}
$$

Note that we have

$$
\sum_{l=0}^{k-1}(1 - \frac{1}{k})^l = \frac{1 - (1 - \frac{1}{k})^k}{1 - (1 - \frac{1}{k})} = k\left(1 - (1 - \frac{1}{k})^k\right).
$$

Hence

$$
\begin{aligned}
f(S^k) &\geq \left[\frac{1}{k}f(S^*) - 2\,\mathrm{rad} - \frac{4d}{m}\right]k\left(1 - (1 - \frac{1}{k})^k\right) \\
&= \left(1 - (1 - \frac{1}{k})^k\right)f(S^*) - \left(2k\,\mathrm{rad} - \frac{4kd}{m}\right)\left(1 - (1 - \frac{1}{k})^k\right) \\
&\geq \left(1 - (1 - \frac{1}{k})^k\right)f(S^*) - 2k\,\mathrm{rad} - \frac{4kd}{m}.
\end{aligned}
$$

Using the well known inequality $(1 - \frac{1}{k})^k \leq \frac{1}{e}$, we get

$$
f(S^k) \geq \left(1 - \frac{1}{e}\right)f(S^*) - 2k\,\mathrm{rad} - \frac{4kd}{m}. \qquad \square
$$

**Lemma 4.** *For all $d > 0$, we have*

$$
\mathbb{E}(\mathcal{R}|\mathcal{E} \cap \mathcal{E}_d') \leq mnk + 2kT\,\mathrm{rad} + \frac{4kTd}{m}.
$$

*Proof.* Let $\mathcal{R} = \mathcal{R}_{\mathrm{exploration}} + \mathcal{R}_{\mathrm{exploitation}}$. The exploration phase is at most $mnk$ steps, therefore we always have

$$
\mathcal{R}_{\mathrm{exploration}} \leq mnk.
$$

At each time-step in the exploitation phase, $(1 - \frac{1}{e})f(S^*) - f(S^k)$ to is added to the expected regret. Hence we have

$$
\begin{aligned}
\mathbb{E}(\mathcal{R}|\mathcal{E} \cap \mathcal{E}_d') &= \mathbb{E}(\mathcal{R}_{\mathrm{exploration}}|\mathcal{E} \cap \mathcal{E}_d') + \mathbb{E}(\mathcal{R}_{\mathrm{exploitation}}|\mathcal{E} \cap \mathcal{E}_d') \\
&\leq mnk + T\left[(1 - \frac{1}{e})f(S^*) - f(S^k)\right] \\
&\leq mnk + 2kT\,\mathrm{rad} + \frac{4kTd}{m},
\end{aligned}
$$

where we used Corollary 1 in the last inequality. $\qquad \square$

**Theorem 8** (Theorem 1 in the main text)**.** *For all $d > 0$, we have*

$$\mathbb{E}(\mathcal{R}) \le mnk + 2kT \operatorname{rad} + \frac{4kTd}{m} + 2nkT \exp(-2m \operatorname{rad}^2) + T(1 - \mathbb{P}(\mathcal{E}'_d)).$$

*Proof.* Using Lemmas 4 and 2, we have

$$
\begin{aligned}
\mathbb{E}(\mathcal{R}) &= \mathbb{E}(\mathcal{R}|\mathcal{E} \cap \mathcal{E}'_d)\mathbb{P}(\mathcal{E} \cap \mathcal{E}'_d) + \mathbb{E}(\mathcal{R}|(\mathcal{E} \cap \mathcal{E}'_d)^c)\mathbb{P}((\mathcal{E} \cap \mathcal{E}'_d)^c) \\
&\le \mathbb{E}(\mathcal{R}|\mathcal{E} \cap \mathcal{E}'_d) + T\mathbb{P}((\mathcal{E} \cap \mathcal{E}'_d)^c) \\
&= \mathbb{E}(\mathcal{R}|\mathcal{E} \cap \mathcal{E}'_d) + T\mathbb{P}(\mathcal{E}^c \cup (\mathcal{E}'_d)^c) \\
&\le \mathbb{E}(\mathcal{R}|\mathcal{E} \cap \mathcal{E}'_d) + T\mathbb{P}(\mathcal{E}^c) + T\mathbb{P}((\mathcal{E}'_d)^c) \\
&\le (mnk + 2kT \operatorname{rad} + \frac{4kTd}{m}) + 2nkT \exp(-2m \operatorname{rad}^2) + T(1 - \mathbb{P}(\mathcal{E}'_d)). \qquad \square
\end{aligned}
$$

## D  Uniformly Bounded Delay

**Lemma 5.** *If delay is uniformly bounded by $d$, then $\mathbb{P}(\mathcal{E}'_d) = 1$.*

*Proof.* For all $t \le 0$, let $F_t = 0$ and let $\delta_t$ be any distribution over non-negative integers. We have

$$
\begin{aligned}
\left| \sum_{t=t_{i,a}}^{t'_{i,a}} X_t - \sum_{t=t_{i,a}}^{t'_{i,a}} F_t \right| &= \left| \sum_{j=t_{i,a}-d}^{t'_{i,a}} F_j \delta_j(\{t_{i,a} - j \le x \le t'_{i,a} - j\}) - \sum_{t=t_{i,a}}^{t'_{i,a}} F_t \right| \\
&= \left| \sum_{j=t_{i,a}-d}^{t_{i,a}-1} F_j \delta_j(\{t_{i,a} - j \le x \le t'_{i,a} - j\}) + \sum_{j=t_{i,a}}^{t'_{i,a}} F_j \delta_j(\{x \le t'_{i,a} - j\}) - \sum_{t=t_{i,a}}^{t'_{i,a}} F_t \right| \\
&= \left| \sum_{j=t_{i,a}-d}^{t_{i,a}-1} F_j \delta_j(\{t_{i,a} - j \le x \le t'_{i,a} - j\}) - \sum_{j=t_{i,a}}^{t'_{i,a}} F_j \delta_j(\{x > t'_{i,a} - j\}) \right| \\
&\le \sum_{j=t_{i,a}-d}^{t_{i,a}-1} \delta_j(\{t_{i,a} - j \le x \le t'_{i,a} - j\}) + \sum_{j=t_{i,a}}^{t'_{i,a}} \delta_j(\{x > t'_{i,a} - j\}) \\
&\le d + \sum_{j=t_{i,a}}^{t'_{i,a}} \delta_j(\{x > t'_{i,a} - j\}).
\end{aligned}
$$

Note that for $j \le t_{i,a} + m - d - 1$, we have

$$\delta_j(\{x > t'_{i,a} - j\}) = \delta_j(\{x > t_{i,a} + m - 1 - j\}) \le \delta_j(\{x > d\}) = 0.$$

Therefore we have

$$
\begin{aligned}
\sum_{j=t_{i,a}}^{t'_{i,a}} \delta_j(\{x > t'_{i,a} - j\}) &= \sum_{j=\max\{t_{i,a}, t_{i,a}+m-d\}}^{t_{i,a}+m-1} \delta_j(\{x > t'_{i,a} - j\}) \\
&\le (t_{i,a} + m - 1) - \max\{t_{i,a}, t_{i,a} + m - d\} + 1 \\
&= \min\{m, d\} \le d.
\end{aligned}
$$

Hence

$$|\bar{X}_{i,a} - \bar{F}_{i,a}| = \frac{1}{m} \left| \sum_{t=t_{i,a}}^{t'_{i,a}} X_t - \sum_{t=t_{i,a}}^{t'_{i,a}} F_t \right| \le \frac{2d}{m}. \qquad \square$$

**Theorem 9** (Theorem 2 in the main text). *If the delay is uniformly bounded by d, then we have*

$$\mathbb{E}(\mathcal{R}) = O(kn^{1/3}T^{2/3}(\log(T))^{1/2}) + O(kn^{2/3}T^{1/3}d).$$

*Proof.* Using Theorem 1 and Lemma 5, we see that

$$\mathbb{E}(\mathcal{R}) \le mnk + 2kT \operatorname{rad} + \frac{4kTd}{m} + 2nkT \exp(-2m \operatorname{rad}^2).$$

Let $\operatorname{rad} := \sqrt{\frac{\log(T)}{m}}$. Then

$$\exp(-2m \operatorname{rad}^2) = T^{-2},$$

and

$$\begin{aligned}
\mathbb{E}(\mathcal{R}) &\le mnk + 2kT \operatorname{rad} + \frac{4kTd}{m} + 2nkT \exp(-2m \operatorname{rad}^2) \\
&= mnk + 2kT\sqrt{\frac{\log(T)}{m}} + \frac{4kTd}{m} + 2nk/T \\
&\le mnk + 2kT\sqrt{\frac{\log(T)}{m}} + \frac{4kTd}{m} + 2k
\end{aligned}$$

where we used $T \ge n$ in the last inequality. Since $m = \lceil (T/n)^{2/3} \rceil$, we have $(T/n)^{2/3} \le m \le (T/n)^{2/3} + 1$. Therefore

$$\begin{aligned}
\mathbb{E}(\mathcal{R}) &\le mnk + 2kT\sqrt{\frac{\log(T)}{m}} + \frac{4kTd}{m} + 2k \\
&\le ((T/n)^{2/3} + 1)nk + 2kT\sqrt{\log(T)/(T/n)^{2/3}} + \frac{4kTd}{(T/n)^{2/3}} + 2k \\
&= kn^{1/3}T^{2/3} + nk + 2kn^{1/3}T^{2/3}(\log(T))^{1/2}) + 4kn^{2/3}T^{1/3}d + 2k \\
&\le 4kn^{1/3}T^{2/3}(\log(T))^{1/2}) + 4kn^{2/3}T^{1/3}d + 2k \\
&= O(kn^{1/3}T^{2/3}(\log(T))^{1/2}) + O(kn^{2/3}T^{1/3}d). \qquad \square
\end{aligned}$$

## E   Unbounded Stochastic Independent Delay

**Lemma 6.** *If $(\Delta_j)_{j=1}^{\infty}$ is independent and $(\mathbb{E}_{\mathcal{T}}(\Delta_j))_{j=1}^{\infty}$ is tight, then we have*

$$\mathbb{P}(\mathcal{E}_d') \ge 1 - nk \exp\left(-\frac{\lambda^2}{2(\mathbb{E}(\tau) + \lambda/3)}\right),$$

*where $d > 0$ is a real number, $\tau$ is a tail upper bound for the family $(\mathbb{E}_{\mathcal{T}}(\Delta_j))_{j=1}^{\infty}$ and $\lambda = \max\{0, d - \mathbb{E}(\tau)\}$.*

*Proof.* If $\lambda = 0$, then the statement is trivially true. So we will assume that $\lambda > 0$ and $d = \mathbb{E}(\tau) + \lambda$. Define

$$C_{i,a} = \sum_{j=1}^{t_{i,a}'} \Delta_j(\{x > t_{i,a}' - j\}).$$

Using the fact that $\tau$ is an upper tail bound for $(\mathbb{E}_{\mathcal{T}}(\Delta_j))_{j=1}^{\infty}$, we can see that

$$\begin{aligned}
\mathbb{E}(C_{i,a}) &= \sum_{j=1}^{t_{i,a}'} \mathbb{E}(\Delta_j(\{x > t_{i,a}' - j\})) \le \sum_{j=1}^{t_{i,a}'} \tau(\{x > t_{i,a}' - j\}) \\
&= \sum_{j=1}^{t_{i,a}'} \tau(\{x \ge j\}) \le \sum_{j=0}^{\infty} \tau(\{x \ge j\}) = \mathbb{E}(\tau).
\end{aligned}$$

Using Bernstein's inequality, we have

$$
\begin{aligned}
\mathbb{P}(C_{i,a} > d) &= \mathbb{P}(C_{i,a} > \mathbb{E}(\tau) + \lambda) \\
&\le \mathbb{P}(C_{i,a} > \mathbb{E}(C_{i,a}) + \lambda) \\
&\le \exp\left(-\frac{\lambda^2}{2(\sum_{j=1}^{t'_{i,a}} \mathbb{E}(\Delta_j(\{x > t'_{i,a} - j\})^2) + \lambda/3)}\right) \\
&\le \exp\left(-\frac{\lambda^2}{2(\sum_{j=1}^{t'_{i,a}} \mathbb{E}(\Delta_j(\{x > t'_{i,a} - j\})) + \lambda/3)}\right) \\
&= \exp\left(-\frac{\lambda^2}{2(\mathbb{E}(C_{i,a}) + \lambda/3)}\right) \\
&\le \exp\left(-\frac{\lambda^2}{2(\mathbb{E}(\tau) + \lambda/3)}\right).
\end{aligned}
$$

We have $X_t = \sum_{j=1}^{t} F_j(S_j)\Delta_j(t-j)$. Therefore

$$
\begin{aligned}
m|\bar{X}_{i,a} - \bar{F}_{i,a}| &= \left|\sum_{t=t_{i,a}}^{t'_{i,a}} X_t - \sum_{t=t_{i,a}}^{t'_{i,a}} F_t\right| \\
&= \left|\sum_{j=1}^{t'_{i,a}} F_j\Delta_j(\{t_{i,a} - j \le x \le t'_{i,a} - j\}) - \sum_{t=t_{i,a}}^{t'_{i,a}} F_t\right| \\
&= \left|\sum_{j=1}^{t_{i,a}-1} F_j\Delta_j(\{t_{i,a} - j \le x \le t'_{i,a} - j\}) + \sum_{j=t_{i,a}}^{t'_{i,a}} F_j\Delta_j(\{x \le t'_{i,a} - j\}) - \sum_{t=t_{i,a}}^{t'_{i,a}} F_t\right| \\
&= \left|\sum_{j=1}^{t_{i,a}-1} F_j\Delta_j(\{t_{i,a} - j \le x \le t'_{i,a} - j\}) - \sum_{j=t_{i,a}}^{t'_{i,a}} F_j\Delta_j(\{x > t'_{i,a} - j\})\right| \\
&\le \sum_{j=1}^{t_{i,a}-1} \Delta_j(\{t_{i,a} - j \le x \le t'_{i,a} - j\}) + \sum_{j=t_{i,a}}^{t'_{i,a}} \Delta_j(\{x > t'_{i,a} - j\}) \\
&\le \sum_{j=1}^{t_{i,a}-1} \Delta_j(\{x \ge t_{i,a} - j\}) + \sum_{j=1}^{t'_{i,a}} \Delta_j(\{x > t'_{i,a} - j\}) \\
&= \sum_{j=1}^{t_{i,a}-1} \Delta_j(\{x > t_{i,a} - 1 - j\}) + C_{i,a}.
\end{aligned}
$$

If $t_{i,a} = 1$, then the first sum will be zero and we have

$$
m|\bar{X}_{i,a} - \bar{F}_{i,a}| \le C_{i,a}.
$$

Otherwise, there exists $(i', a')$ such that $t'_{i',a'} = t_{i,a} - 1$ and

$$
m|\bar{X}_{i,a} - \bar{F}_{i,a}| \le C_{i',a'} + C_{i,a}.
$$

Let $\mathcal{E}^*_{i,a}$ be the event that $C_{i,a} \le d$ and define

$$
\mathcal{E}^* := \bigcap_{i,a} \mathcal{E}^*_{i,a}.
$$

Our discussion above shows that we have

$$\mathbb{P}(\mathcal{E}'_d) \geq \mathbb{P}(\mathcal{E}^*).$$

On the other hand, we have

$$
\begin{aligned}
\mathbb{P}(\mathcal{E}^*) &= \mathbb{P}\left(\bigcap_{i,a} \mathcal{E}^*_{i,a}\right) \\
&= 1 - \mathbb{P}\left(\bigcup_{i,a} (\mathcal{E}^*_{i,a})^c\right) \\
&= 1 - \mathbb{P}\left(\bigcup_{i,a} \{C_{i,a} > d\}\right) \\
&\geq 1 - \sum_{i,a} \mathbb{P}\left(\{C_{i,a} > d\}\right) \\
&\geq 1 - \sum_{i,a} \exp\left(-\frac{\lambda^2}{2(\mathbb{E}(\tau) + \lambda/3)}\right) \\
&\geq 1 - nk \exp\left(-\frac{\lambda^2}{2(\mathbb{E}(\tau) + \lambda/3)}\right),
\end{aligned}
$$

which completes the proof. $\qquad\square$

**Theorem 10** (Theorem 3 in the main text)**.** *If the delay sequence is stochastic, then we have*

$$\mathbb{E}(\mathcal{R}) = O(kn^{1/3}T^{2/3}\log(T)) + O(kn^{2/3}T^{1/3}\mathbb{E}(\tau)),$$

*where $\tau$ is an upper tail bound for $(\mathbb{E}_{\mathcal{T}}(\Delta_t))_{t=1}^\infty$.*

*Proof.* Let $\mathrm{rad} := \sqrt{\frac{\log(T)}{m}}$. Then, using $T \geq n$, we have

$$2nkT \exp(-2m\,\mathrm{rad}^2) = 2nkT \exp(-2\log(T)) = 2nkT^{-1} \leq 2k.$$

We choose $d := \mathbb{E}(\tau) + \max\{6\mathbb{E}(\tau), 2\log(T)\}$ and $\lambda = d - \mathbb{E}(\tau) = \max\{6\mathbb{E}(\tau), 2\log(T)\}$. Then we have

$$\exp\left(-\frac{\lambda^2}{2(\mathbb{E}(\tau) + \lambda/3)}\right) \leq \exp\left(-\frac{\lambda^2}{2(\lambda/6 + \lambda/3)}\right) = \exp(-\lambda) \leq \exp(-2\log(T)) = T^{-2}.$$

Therefore

$$nkT \exp\left(-\frac{\lambda^2}{2(\mathbb{E}(\tau) + \lambda/3)}\right) \leq nkT^{-1} \leq k.$$

So, using Theorem 1 and Lemma 6, we have

$$
\begin{aligned}
\mathbb{E}(\mathcal{R}) &\leq mnk + 2kT\,\mathrm{rad} + \frac{4kTd}{m} + 2nkT\exp(-2m\,\mathrm{rad}^2) + T(1 - \mathbb{P}(\mathcal{E}'_d)) \\
&\leq mnk + 2kT\,\mathrm{rad} + \frac{4kTd}{m} + 2nkT\exp(-2m\,\mathrm{rad}^2) + nkT\exp\left(-\frac{\lambda^2}{2(\mathbb{E}(\tau) + \lambda/3)}\right) \\
&\leq mnk + 2kT\,\mathrm{rad} + \frac{4kTd}{m} + 3k \\
&= mnk + 2kT\sqrt{\frac{\log(T)}{m}} + 4k\frac{T}{m}\left(\mathbb{E}(\tau) + \max\{6\mathbb{E}(\tau), 2\log(T)\}\right) + 3k \\
&\leq mnk + 2kT\sqrt{\frac{\log(T)}{m}} + 4k\frac{T}{m}\left(7\mathbb{E}(\tau) + 2\log(T)\right) + 3k.
\end{aligned}
$$

Since $m = \lceil (T/n)^{2/3} \rceil$, we have $(T/n)^{2/3} \le m \le (T/n)^{2/3} + 1$. Therefore

$$
\begin{aligned}
\mathbb{E}(\mathcal{R}) &\le mnk + 2kT\sqrt{\frac{\log(T)}{m}} + \frac{4kT}{m}\left(7\mathbb{E}(\tau) + 2\log(T)\right) + 3k \\
&\le ((T/n)^{2/3} + 1)nk + 2kT\sqrt{\frac{\log(T)}{(T/n)^{2/3}}} + \frac{4kT}{(T/n)^{2/3}}\left(7\mathbb{E}(\tau) + 2\log(T)\right) + 3k \\
&= kn^{1/3}T^{2/3} + kn + 2kn^{1/3}T^{2/3}\log(T)^{1/2} + 28kn^{2/3}T^{1/3}\mathbb{E}(\tau) + 8kn^{2/3}T^{1/3}\log(T) + 3k \\
&\le 12kn^{1/3}T^{2/3}\log(T) + 28kn^{2/3}T^{1/3}\mathbb{E}(\tau) + 3k \\
&= O(kn^{1/3}T^{2/3}\log(T)) + O(kn^{2/3}T^{1/3}\mathbb{E}(\tau)). \qquad \square
\end{aligned}
$$

## F  Unbounded Stochastic Conditionally Independent Delay

**Lemma 7.** *If $(\Delta_{j,S})_{j=1}^{\infty}$ is pairwise independent for all $S \in \mathcal{S}$ and $\{\mathbb{E}(\Delta_{j,S})\}_{j\ge 1, S\in\mathcal{S}}$ is tight, then we have*

$$
\mathbb{P}(\mathcal{E}_d') \ge 1 - nk\exp\left(-\frac{\lambda^2}{2(\mathbb{E}(\tau) + \lambda/3)}\right),
$$

*where $d > 0$ is a real number, $\tau$ is a tail upper bound for the family $\{\mathbb{E}_{\mathcal{T}}(\Delta_{j,S})\}_{j\ge 1, S\in\mathcal{S}}$ and $\lambda = \max\left\{0, \frac{2d}{nk} - \mathbb{E}(\tau)\right\}$.*

*Proof.* If $\lambda = 0$, then the statement is trivially true. So we will assume that $\lambda > 0$ and $d = \frac{nk}{2}(\mathbb{E}(\tau) + \lambda)$. Define

$$
C_{i,a}' = \sum_{j=t_{i,a}}^{t_{i,a}'} \Delta_j(\{x > t_{i,a}' - j\}).
$$

Note that the sum is only over the time-steps where the action $S^{i-1} \cup \{a\}$ is taken. Therefore $C_{i,a}'$ is the sum of $m$ independent term. Using the fact that $\tau$ is an upper tail bound for $\{\mathbb{E}_{\mathcal{T}}(\Delta_{j,S})\}_{j\ge 1, S\in\mathcal{S}}$, we can see that

$$
\begin{aligned}
\mathbb{E}(C_{i,a}') &= \sum_{j=t_{i,a}}^{t_{i,a}'} \mathbb{E}(\Delta_j(\{x > t_{i,a}' - j\})) \le \sum_{j=t_{i,a}}^{t_{i,a}'} \tau(\{x > t_{i,a}' - j\}) \\
&\le \sum_{j=1}^{t_{i,a}'} \tau(\{x > t_{i,a}' - j\}) = \sum_{j=1}^{t_{i,a}'} \tau(\{x \ge j\}) \le \sum_{j=0}^{\infty} \tau(\{x \ge j\}) = \mathbb{E}(\tau).
\end{aligned}
$$

Using Bernstein's inequality, we have

$$
\begin{aligned}
\mathbb{P}\left(C_{i,a}' > \frac{2d}{nk}\right) &= \mathbb{P}(C_{i,a}' > \mathbb{E}(\tau) + \lambda) \\
&\le \mathbb{P}(C_{i,a}' > \mathbb{E}(C_{i,a}') + \lambda) \\
&\le \exp\left(-\frac{\lambda^2}{2(\sum_{j=t_{i,a}}^{t_{i,a}'} \mathbb{E}(\Delta_j(\{x > t_{i,a}' - j\})^2) + \lambda/3)}\right) \\
&\le \exp\left(-\frac{\lambda^2}{2(\sum_{j=t_{i,a}}^{t_{i,a}'} \mathbb{E}(\Delta_j(\{x > t_{i,a}' - j\})) + \lambda/3)}\right) \\
&= \exp\left(-\frac{\lambda^2}{2(\mathbb{E}(C_{i,a}') + \lambda/3)}\right) \\
&\le \exp\left(-\frac{\lambda^2}{2(\mathbb{E}(\tau) + \lambda/3)}\right).
\end{aligned}
$$

We have $X_t = \sum_{j=1}^{t} F_j(S_j)\Delta_j(t-j)$. Therefore

$$
\begin{aligned}
m|\bar{X}_{i,a} - \bar{F}_{i,a}| &= \left| \sum_{t=t_{i,a}}^{t'_{i,a}} X_t - \sum_{t=t_{i,a}}^{t'_{i,a}} F_t \right| \\
&= \left| \sum_{j=1}^{t'_{i,a}} F_j \Delta_j(\{t_{i,a} - j \le x \le t'_{i,a} - j\}) - \sum_{t=t_{i,a}}^{t'_{i,a}} F_t \right| \\
&= \left| \sum_{j=1}^{t_{i,a}-1} F_j \Delta_j(\{t_{i,a} - j \le x \le t'_{i,a} - j\}) + \sum_{j=t_{i,a}}^{t'_{i,a}} F_j \Delta_j(\{x \le t'_{i,a} - j\}) - \sum_{t=t_{i,a}}^{t'_{i,a}} F_t \right| \\
&= \left| \sum_{j=1}^{t_{i,a}-1} F_j \Delta_j(\{t_{i,a} - j \le x \le t'_{i,a} - j\}) - \sum_{j=t_{i,a}}^{t'_{i,a}} F_j \Delta_j(\{x > t'_{i,a} - j\}) \right| \\
&\le \sum_{j=1}^{t_{i,a}-1} \Delta_j(\{t_{i,a} - j \le x \le t'_{i,a} - j\}) + \sum_{j=t_{i,a}}^{t'_{i,a}} \Delta_j(\{x > t'_{i,a} - j\}) \\
&\le \sum_{j=1}^{t_{i,a}-1} \Delta_j(\{x > t_{i,a} - 1 - j\}) + C'_{i,a}.
\end{aligned}
$$

Define

$$
I_{i,a} = \{(i', a') \mid t_{i',a'} < t_{i,a}\}.
$$

Then we have

$$
\begin{aligned}
\sum_{j=1}^{t_{i,a}-1} \Delta_j(\{x > t_{i,a} - 1 - j\}) &= \sum_{(i',a') \in I_{i,a}} \sum_{j=t_{i',a'}}^{t'_{i',a'}} \Delta_j(\{x > t_{i,a} - 1 - j\}) \\
&\le \sum_{(i',a') \in I_{i,a}} \sum_{j=t_{i',a'}}^{t'_{i',a'}} \Delta_j(\{x > t'_{i',a'} - j\}) = \sum_{(i',a') \in I_{i,a}} C'_{i',a'}.
\end{aligned}
$$

Therefore, we have

$$
m|\bar{X}_{i,a} - \bar{F}_{i,a}| \le \sum_{i,a} C'_{i,a} \le nk \max_{i,a}\{C'_{i,a}\}.
$$

Let $\mathcal{E}^{**}_{i,a}$ be the event that $C_{i,a} \le \frac{2d}{nk}$ and define

$$
\mathcal{E}^{**} := \bigcap_{i,a} \mathcal{E}^{**}_{i,a}.
$$

Our discussion above shows that we have

$$
\mathbb{P}(\mathcal{E}'_d) \ge \mathbb{P}(\mathcal{E}^{**}).
$$

On the other hand, we have

$$\mathbb{P}(\mathcal{E}^{**}) = \mathbb{P}\left(\bigcap_{i,a} \mathcal{E}_{i,a}^{**}\right)$$

$$= 1 - \mathbb{P}\left(\bigcup_{i,a}(\mathcal{E}_{i,a}^{**})^c\right)$$

$$= 1 - \mathbb{P}\left(\bigcup_{i,a}\left\{C_{i,a}' > \frac{2d}{nk}\right\}\right)$$

$$\geq 1 - \sum_{i,a}\mathbb{P}\left(\left\{C_{i,a}' > \frac{2d}{nk}\right\}\right)$$

$$\geq 1 - \sum_{i,a}\exp\left(-\frac{\lambda^2}{2(\mathbb{E}(\tau) + \lambda/3)}\right)$$

$$\geq 1 - nk\exp\left(-\frac{\lambda^2}{2(\mathbb{E}(\tau) + \lambda/3)}\right),$$

which completes the proof. $\qquad\square$

**Theorem 11** (Theorem 4 in the main text)**.** *If the delay sequence is stochastic and conditionally independent, then we have*

$$\mathbb{E}(\mathcal{R}) = O(kn^{1/3}T^{2/3}(\log(T))^{1/2} + k^2 n^{5/3}T^{1/3}\log(T)) + O(k^2 n^{5/3}T^{1/3}\mathbb{E}(\tau))$$
$$= O(k^2 n^{4/3}T^{2/3}\log(T)) + O(k^2 n^{5/3}T^{1/3}\mathbb{E}(\tau))$$

*where $\tau$ is an upper tail bound for $(\mathbb{E}_{\mathcal{T}}(\Delta_t))_{t=1}^{\infty}$.*

*Proof.* Let $\mathrm{rad} := \sqrt{\frac{\log(T)}{m}}$. Then, using $T \geq n$, we have

$$2nkT\exp(-2m\,\mathrm{rad}^2) = 2nkT\exp(-2\log(T)) = 2nkT^{-1} \leq 2k.$$

We choose $d := \frac{nk}{2}(\mathbb{E}(\tau) + \max\{6\mathbb{E}(\tau), 2\log(T)\})$ and $\lambda = \frac{2d}{nk} - \mathbb{E}(\tau) = \max\{6\mathbb{E}(\tau), 2\log(T)\}$. Then we have

$$\exp\left(-\frac{\lambda^2}{2(\mathbb{E}(\tau) + \lambda/3)}\right) \leq \exp\left(-\frac{\lambda^2}{2(\lambda/6 + \lambda/3)}\right) = \exp(-\lambda) \leq \exp(-2\log(T)) = T^{-2}.$$

Therefore

$$nkT\exp\left(-\frac{\lambda^2}{2(\mathbb{E}(\tau) + \lambda/3)}\right) \leq nkT^{-1} \leq k.$$

So, using Theorem 1 and Lemma 7, we have

$$\mathbb{E}(\mathcal{R}) \leq mnk + 2kT\,\mathrm{rad} + \frac{4kTd}{m} + 2nkT\exp(-2m\,\mathrm{rad}^2) + T(1 - \mathbb{P}(\mathcal{E}_d'))$$

$$\leq mnk + 2kT\,\mathrm{rad} + \frac{4kTd}{m} + 2nkT\exp(-2m\,\mathrm{rad}^2) + nkT\exp\left(-\frac{\lambda^2}{2(\mathbb{E}(\tau) + \lambda/3)}\right)$$

$$\leq mnk + 2kT\,\mathrm{rad} + \frac{4kTd}{m} + 3k$$

$$= mnk + 2kT\sqrt{\frac{\log(T)}{m}} + \frac{2nk^2T}{m}\left(\mathbb{E}(\tau) + \max\{6\mathbb{E}(\tau), 2\log(T)\}\right) + 3k$$

$$\leq mnk + 2kT\sqrt{\frac{\log(T)}{m}} + \frac{2nk^2T}{m}\left(7\mathbb{E}(\tau) + 2\log(T)\right) + 3k.$$

Since $m = \lceil (T/n)^{2/3} \rceil$, we have $(T/n)^{2/3} \leq m \leq (T/n)^{2/3} + 1$. Therefore

$$
\begin{aligned}
\mathbb{E}(\mathcal{R}) &\leq mnk + 2kT\sqrt{\frac{\log(T)}{m}} + \frac{2nk^2 T}{m}\left(7\mathbb{E}(\tau) + 2\log(T)\right) + 3k \\
&\leq ((T/n)^{2/3} + 1)nk + 2kT\sqrt{\frac{\log(T)}{(T/n)^{2/3}}} + \frac{2nk^2 T}{(T/n)^{2/3}}\left(7\mathbb{E}(\tau) + 2\log(T)\right) + 3k \\
&= kn^{1/3}T^{2/3} + kn + 2kn^{1/3}T^{2/3}\log(T)^{1/2} + 14k^2 n^{5/3}T^{1/3}\mathbb{E}(\tau) + 4k^2 n^{5/3}T^{1/3}\log(T) + 3k \\
&= O(kn^{1/3}T^{2/3}(\log(T))^{1/2}) + O(k^2 n^{5/3}T^{1/3}\log(T)) + O(k^2 n^{5/3}T^{1/3}\mathbb{E}(\tau)) \\
&= O(k^2 n^{4/3}T^{2/3}\log(T)) + O(k^2 n^{5/3}T^{1/3}\mathbb{E}(\tau)). \qquad \square
\end{aligned}
$$

## G  Extension to general combinatorial bandits

The results of this paper could be generalized to settings beyond monotone submodular bandits with cardinality constraint. As we will see, instead of these assumptions, we only need a setting where we have an algorithm for the offline problem satisfying a specific notion of robustness.

As before, let $\Omega$ be the set of base arms and let $\mathcal{S}$ be a subset of $2^\Omega$. Let $\mathcal{F}$ be a class of functions from $\mathcal{S} \to [0,1]$ where we know that $f \in \mathcal{F}$. We use $S^*$ to denote the optimal value of $f$.

**Definition 2** ((Nie et al., 2023))**.** Let $\mathcal{A}$ be an algorithm for the combinatorial optimization problem of maximizing a function $f : \mathcal{S} \to \mathbb{R}$ over a finite domain $\mathcal{S} \subseteq 2^\Omega$ with the knowledge that $f$ belongs to a known class of functions $\mathcal{F}$. for any function $\hat{f} : \mathcal{S} \to \mathbb{R}$, let $\mathcal{S}_{\mathcal{A},\hat{f}}$ denote the output of $\mathcal{A}$ when it is run with $\hat{f}$ as its value oracle. The algorithm $\mathcal{A}$ called $(\alpha, \delta)$-robust if for any $\epsilon > 0$ and any function $\hat{f}$ such that $|f(S) - \hat{f}(S)| < \epsilon$ for all $S \in \mathcal{S}$, we have

$$
f(S_{\mathcal{A},\hat{f}}) \geq \alpha f(S^*) - \delta\epsilon.
$$

In this setting, $N$ is an upper-bound for the number of $\mathcal{A}$'s queries to the value oracle.

In the previous sections, the set $\mathcal{S}$ was the set of all subsets of $\Omega$ with size at most $k$ and $\mathcal{F}$ was the set of monotone submodular functions on $\mathcal{S}$. Corollary 1 simply states that the greedy algorithm is $(1 - 1/e, 2k)$-robust. If we choose $\mathcal{A}$ to be the offline greedy algorithm, $\alpha = 1 - 1/e$, $\delta = 2k$ and $N = nk$, then Algorithm 2 will reduce to Algorithm 1.

---

**Algorithm 2** C-ETC algorithm ((Nie et al., 2023)

---

**Input:** Set of base arms $\Omega$, horizon $T$, an offline $(\alpha, \delta)$-robust algorithm $\mathcal{A}$, and an upper-bound $N$ on the number of $\mathcal{A}$'s queries to the value oracle
**Assumption:** $N \leq T$
 1: $m \leftarrow \lceil (\delta T/N)^{2/3} \rceil$
 2: **while** $\mathcal{A}$ queries the value of some action $S$ **do**
 3:    Play $S$ arm $m$ times
 4:    Calculate the empirical mean $\bar{x}$
 5:    Return $\bar{x}$ to $\mathcal{A}$
 6: **end while**
 7: **for** remaining time **do**
 8:    Play action $S_\mathcal{A}$ output by algorithm $\mathcal{A}$
 9: **end for**

---

The proof only needs minor changes to adapt for Algorithm 2. Lemma 2 immediately generalizes to

$$
\mathbb{P}(\mathcal{E}) \geq 1 - 2N\exp(-2m\,\mathrm{rad}^2),
$$

where $nk$ is replaced by $N$. Instead of Corollary 1, we need the following statement.

**Corollary 2.** *Under the event $\mathcal{E} \cap \mathcal{E}'_d$, for all $d > 0$, we have*

$$f(S_{\mathcal{A}}) \geq \alpha f(S^*) - \delta \left( \text{rad} + \frac{2d}{m} \right).$$

*Proof.* Consider a time interval of length $m$, namely $t, t+1, \cdots, t+m-1$, where an action $S$ is repeated and the empirical mean $\bar{x}$ is observed. We have

$$m|\bar{x} - f(S)| = \left| \sum_{i=t}^{t+m-1} (x_t - f(S)) \right| \leq \left| \sum_{i=t}^{t+m-1} (x_t - f_t) \right| + \left| \sum_{i=t}^{t+m-1} (f_t - f(S)) \right|.$$

Under the event $\mathcal{E}$, we have

$$\left| \sum_{i=t}^{t+m-1} (f_t - f(S)) \right| \leq m \, \text{rad},$$

and under the event $\mathcal{E}_d$, we have

$$\left| \sum_{i=t}^{t+m-1} (x_t - f_t) \right| \leq 2d.$$

Therefore, we have

$$|\bar{x} - f(S)| \leq \text{rad} + \frac{2d}{m}.$$

Now the claim follows from the definition of $(\alpha, \delta)$-robustness of $\mathcal{A}$. $\qquad\square$

The proofs of Lemma 4 and Theorem 1 could be applied almost verbatim to give us

$$\mathbb{E}(\mathcal{R}_\alpha) \leq mN + \delta T \, \text{rad} + \frac{2\delta T d}{m} + 2NT \exp(-2m \, \text{rad}^2) + T(1 - \mathbb{P}(\mathcal{E}'_d)),$$

for all $d > 0$. The results below follow.

**Theorem 12.** *If the delay is uniformly bounded by $d$, then we have*

$$\mathbb{E}(\mathcal{R}_\alpha) = O(N^{1/3}\delta^{2/3}T^{2/3}(\log(T))^{1/2}) + O(N^{2/3}\delta^{1/3}T^{1/3}d).$$

**Theorem 13.** *If the delay sequence is stochastic, then we have*

$$\mathbb{E}(\mathcal{R}_\alpha) = O(N^{1/3}\delta^{2/3}T^{2/3}\log(T)) + O(N^{2/3}\delta^{1/3}T^{1/3}\mathbb{E}(\tau)),$$

*where $\tau$ is an upper tail bound for $(\mathbb{E}_{\mathcal{T}}(\Delta_t))_{t=1}^\infty$.*

**Theorem 14.** *If the delay sequence is stochastic and conditionally independent, then we have*

$$\mathbb{E}(\mathcal{R}_\alpha) = O(N^{4/3}\delta^{2/3}T^{2/3}\log(T)) + O(N^{5/3}\delta^{1/3}T^{1/3}\mathbb{E}(\tau)),$$

*where $\tau$ is an upper tail bound for $(\mathbb{E}_{\mathcal{T}}(\Delta_t))_{t=1}^\infty$.*

## H   Details on Function, Delay Settings, and Baselines in Evaluations

**Submodular functions:**

(F1) Linear:

Here we assume that $f$ is a linear function of the individual arms. In particular, for a function $g : \Omega \to [0, 1]$, we define

$$f(S) := \frac{1}{k} \sum_{a \in S} g(a).$$

More specifically, we let $n = 20$ and $k = 4$ and choose $g(a)$ uniformly from $[0.1, 0.9]$, for all $a \in \Omega$ and define $F(S) := \frac{1}{k} \sum_{a \in S} g(a) + N^c(0, 0.1)$ where $N^c(0, 0.1)$ is the truncated normal distribution with mean 0 and standard deviation 0.1, truncated to the interval $[-0.1, 1.0]$.

(F2) Weight Cover:

Here we assume that $(C_j)_{j \in J}$ is a partition of $\Omega$ and there is a weight function $w_t : J \to [0, 1]$. Then $f_t(S)$ is the sum of the weights of the the indexes $j$ where $C_j \cap S \neq \emptyset$, divided by $k$. In other words, if $1$ is the indicator function, then

$$f_t(S) := \frac{1}{k} \sum_{j \in J} w_t(j) \, 1_{S \cap C_j \neq \emptyset} \,.$$

More specifically, we let $n = 20$ and $k = 4$. We divide $\Omega$ into 4 categories of sizes $6, 6, 6, 2$ and let $w_t(j) = U([0, j/5])$ be samples uniformly from $[0, j/5]$ for $j \in 1, 2, 3, 4$.

Stochastic set cover may be viewed as a simple model for product recommendation. Assume $n$ is the number of the products and each product belongs to exactly one of $c$ categories. Then the reward will be equal to the sum of the weights of the categories that have been covered by the user divided by $k$.

**Delay settings:**

(D1) No Delay

(D2) (Stochastic Independent Delay) For all $t \geq 1$ and $i \geq 0$, $\Delta_t(i) = (1 - Y_t)Y_t^i$, where $(Y_t)_{t=1}^{\infty}$ is an i.i.d sequence of random variables with the uniform distribution $U([0.5, 0.9])$. The reward for time-step $t$ will be distributed over $[t, \infty)$ according to $\Delta_t$. In other words, at each time-step $t$, the agent plays the action $S_t$, then the environment samples $f_t(S_t)$ according to the distribution of $F_t(S_t)$ and samples $y_t$ according to the distribution $U([0.5, 0.9])$. Then we have $\delta_t(i) = (1 - y_t)y_t^i$ for all $i \geq 0$, which is used in Equation 1 to determine the observation. In this example, the distribution of $\Delta_t$ does not depend on the action chosen by the agent and $(\Delta_t)_{t=1}^{\infty}$ is i.i.d.

(D3) (Stochastic Independent Delay) For all $t \geq 0$, $\Delta_t$ is a distribution over $[10, 30]$ is sampled uniformly from the probability simplex using the flat Dirichlet distribution. The reward for time-step $t$ will be distributed over $[t+10, t+30]$ according to $\Delta_t$. In other words, at each time-step $t$, the agent plays the action $S_t$, then the environment samples $f_t(S_t)$ according to the distribution of $F_t(S_t)$ and samples $(\beta_0, \beta_2, \cdots, \beta_{20})$ from the 20-dimensional probability simplex $\{(z_0, \cdots, z_{20}) \mid z_i \geq 0, \sum z_i = 1\}$, according to the flat Dirichlet distribution. Then we have $\delta_t(i) = \beta_{i-10}$ for all $10 \leq i \leq 30$ and $\delta_t(i) = 0$ otherwise, which is used in Equation 1 to determine the observation. In this example, the distribution of $\Delta_t$ does not depend on the action chosen by the agent and $(\Delta_t)_{t=1}^{\infty}$ is i.i.d.

(D4) (Stochastic Conditionally Independent Delay) For all $t \geq 1$ and $i \geq 0$, we have $\Delta_t(i) = (1 - Y_t)Y_t^i$, where $Y_t = 0.5 + f_t * 0.4 \in [0.5, 0.9]$. The reward for time-step $t$ will be distributed over $[t, \infty)$ according to $\Delta_t$. In other words, at each time-step $t$, the agent plays the action $S_t$, then the environment samples $f_t(S_t)$ according to the distribution of $F_t(S_t)$ and picks $y_t = 0.5 + f_t(S_t) * 0.4$. Then we have $\delta_t(i) = (1-y_t)y_t^i$ for all $i \geq 0$, which is used in Equation 1 to determine the observation. Note that there is no more randomness in delay after the value of $f_t(S_t)$ is samples from $F_t(S_t)$. Also note that the value of $y_t$ depends on the action of the agent. In this example $(\Delta_{t,S})_{t \geq 1}$ is pair-wise independent for any $S \in \mathcal{S}$.

(D5) (Stochastic Conditionally Independent Delay) At each time-step $t$, a number $l_t$ is chosen from $[10, 30]$ according to the following formula.
$$l_t = \lfloor 20f_t \rfloor + 10.$$

The reward for time-step $t$ will be observed at $t + l_t$. More specifically, at each time-step $t$, the agent plays the action $S_t$, then the environment samples $f_t(S_t)$ according to the distribution of $F_t(S_t)$ and picks $l_t = \lfloor 20f_t(S_t) \rfloor + 10$. Finally, we have $\delta_t(i) = \mathbf{1}_{i=l_t}$. In other words, the higher the reward, the more it will be delayed. In this example, delay depends on the action chosen by the agent and $(\Delta_{t,S})_{t \geq 1}$ is pair-wise independent for any $S \in \mathcal{S}$.

(D6) (Adversarial Delay) Let $l_1 = 15$ and for all $t > 1$, define $l_t$ according to the following formula.

$$l_t = \lfloor 20x_{t-1} \rfloor + 10.$$

As in the delay (D5), the value of $l_t$ determines the amount of delay, i.e. $\delta_t(i) = \mathbf{1}_{i=l_t}$. Note that $x_{t-1}$ is the value of the previous observation as described in Equation 1. In other words, the higher the previous observation, the more the current reward will be delayed.

**Baselines:**

We use three algorithms designed for CMAB with full-bandit feedback without delay and and algorithm designed for MAB with composite anonymous feedback as the baseline.

- **CMAB-SM** (Agarwal et al., 2022) This algorithm assumes the expected reward functions are Lipschitz continuous of individual base arm rewards. CMAB-SM has a theoretical 1-regret guarantee of $\tilde{O}(T^{2/3})$ with the assumption that if arm $a$ is better than arm $b$, then replacing $b$ by $a$ in any set not including $a$ will give better reward function.

- **DART** (Agarwal et al., 2021) This algorithm assumes the expected reward functions are Lipschitz continuous of individual base arm rewards and the reward functions have an additional property related to the marginal gains of the base arms. DART has a theoretical 1-regret guarantee of $\tilde{O}(T^{1/2})$ with the assumption that if arm $a$ is better than arm $b$, then replacing $b$ by $a$ in any set not including $a$ will give better reward function.

- **OG$^o$** (Streeter & Golovin, 2008) This algorithm is designed for oblivious adversarial setting with submodular rewards. Therefore the sequence of monotone and submodular functions is fixed in advance. It has an $(1 - 1/e)$-regret guarantee of $\tilde{O}(T^{2/3})$.

- **ARS-UCB** (Wang et al., 2021) This algorithm is designed for MAB with composite anonymous delayed feedback. The delay model is a special case of *unbounded stochastic conditionally independent composite anonymous feedback delay* that we described. However, they assume that $\Delta_{t,S}$ does not depend on time. For our experiments, we use all subsets of $\Omega$ of size $k$ as the set of arms. ARS-UCB has a 1-regret guarantee of $\tilde{O}(\binom{n}{k}^{1/2}T^{1/2})$ plus a constant term that depends on delay and the number of its arms.

# I Experiments with added regret

In Figure 2, we have considered the same functions and delay types as before. After fixing the function and a delay type, we ran each experiment with and without delay 10 times and plotted the added regret when delay is present. In these experiments, we see that the added regret for ETCG is consistently relatively low with low variance. We note that one should be careful when interpreting these plots, since the regret bounds are simply upper bounds and therefore the values shown here are the difference of two values that are bounded from above. Specifically, having upper bounds $\mathcal{R}^{\text{delay}} \leq aT^{2/3} + \nu T^{1/3}$ and $\mathcal{R}^{\text{no-delay}} \leq aT^{2/3}$ do not imply $\mathcal{R}^{\text{delay}} - \mathcal{R}^{\text{no-delay}} \leq \nu T^{1/3}$.

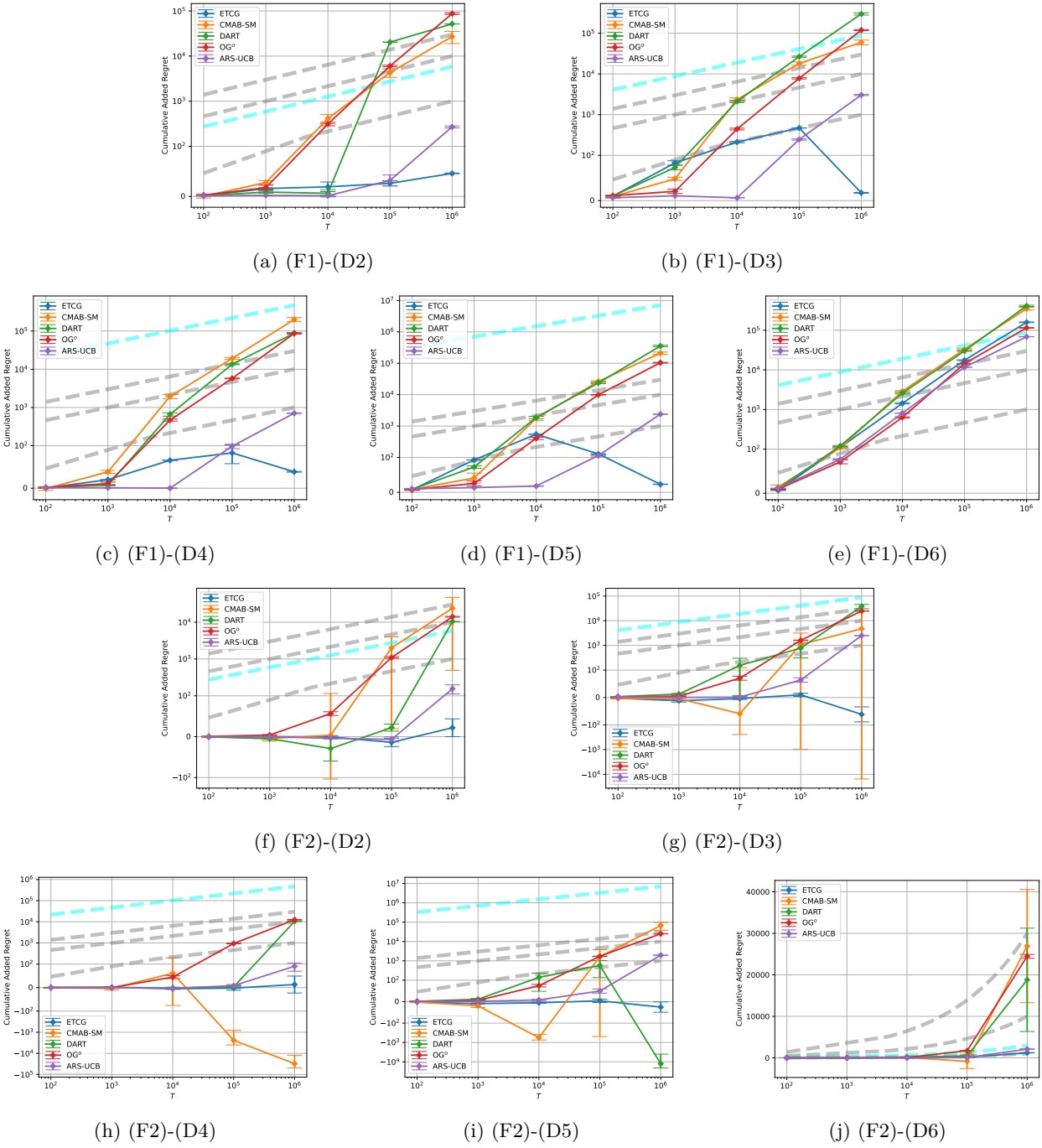

Figure 2: This plot shows the average added cumulative regret over horizon for each setting in the symlog-log scale over 10 runs. The scale of the y-axis is linear for $|y| \leq 100$ and logarithmic for $|y| > 100$. The gray dashed lines are $y = aT^{1/3}$ for $a \in \{10, 100, 300\}$. The cyan dashed lines are $y = \nu T^{1/3}$ where $\nu$ is the corresponding delay coefficient appearing in the regret bounds in Theorems 2, 3 and 4.

