# OpenReview forum: "Stochastic Submodular Bandits with Delayed Composite Anonymous Bandit Feedback"
_TMLR — Rejected by TMLR_

### Review · Reviewer_yFzK · 2023-09-09

**Summary Of Contributions:**

The paper studies the combinatorial multi-armed bandit problem with stochastic rewards and delayed full-bandit feedback. It considers 3 delay models, namely independent stochastic, conditionally independent stochastic, and adversarial delays. It then bounds the regret of explore-then-commit-greedy algorithm from Nie et al., 2022, under the 3 different delay models. The paper generalized their analysis to a broader range of delayed bandits as long as there exists an algorithm for the offline problem satisfying a certain error lower bound.
The experimental results complement the theoretical analysis and compares few benchmark algorithms together under different conditions delay and reward assumptions.

**Audience:**

Yes

**Broader Impact Concerns:**

Not needed.

**Claims And Evidence:**

Yes

**Requested Changes:**

Please see the Weaknesses part above. The presentation should improve, this is critical.

**Strengths And Weaknesses:**

Strengths:
1. The paper has comprehensive and rigorous analysis of several different delay models and bounds the regret of the algorithm under each setting in detail.
2. The paper is well-motivated: the applications are instantiated, and it is clear why the paper is considering such a problem. Prior works are studied and discussed very well.

Weaknesses:
1. Considering the paper is considering a new problem different than Nie et al., 2022, it could consider proposing a new algorithm that fits the problem at hand. The paper simply analyses the existing ETCG algorithm. It is not clear if this algorithm achieves any optimality or how this algorithm compares to the others. The experimental results show the performance of different algorithms are very similar.
2. The representation could improve:
(i) The notation gets overloaded and sometimes ambiguous: for example, $f$ is both used for the observation and also the expectation (see the beginning of Section 2). $m$ is only implicitly in the pesudo-code in Algorithm 1. $\tau$ after Lemma 1 is not clearly defined and it takes an effort to understand what it symbolizes.
(ii) The proofs are sketched but the results (bounds) are not discussed. For instance, it could be compared to other existing delayed bandit (not necessarily combinatorial) bounds in detail.

---

> ### Author Response · Authors · 2023-10-06
>
> Thanks a lot for your comments and for appreciating our results.
>
> 1. The problem of aggregated anonymous delay was first studied by (Pike-Burke et al., 2018). This notion was generalized to composite anonymous delay in (Cesa-Bianchi et al., 2018) in the adversarial feedback setting. Later (Garg \& Akash, 2019) and (Wang et al., 2021) studied the same delay type in the stochastic feedback setting. Our work is the study of the same type of delay for combinatorial bandits in general and combinatorial submodular bandits in particular.
> Many different types of delay has been studied in the literature but to the best of our knowledge, the composite anonymous delay is the least informative type.
> We note in all of these works addressing composite anonymous delay, including ours, a key idea is to repeat actions enough times so that we can extract meaningful information. This is not always necessary in other types of delay. In particular, if delay is not anonymous, there is no need to repeat actions since we will eventually know the reward for each action.
> Other algorithms discussed in the paper for combinatoral bandits do not repeat actions and therefore there is little hope of them achieving desirable results in the presence of composite anonymous delay.
> Moreover, it should be noted that other algorithms do not address the general problem of stochastic submodular combinatoral bandit and we have simply included them to have something to compare to.
> As described in Appendix H, other baselines are either for adversarial setting (e.g. OG$^o$), which is different than stochastic, or have stronger assumptions (e.g. CMAB-SM and DART), or are not designed for combinatoral setting (e.g. ARS-UCB). Thus, we note that ETCG is the only algorithm with regret guarantees for submodular bandits with full-bandit feedback in the absence of delay.
> 2. (i) We note that we have used lower case variables to denote deterministic values and functions and upper case letters to denote random variables (with the exception of delay distributions $\delta_t$ which are, for the most part, treated as determinisitc functions over non-negative integers).
> We have also tried to have different notations for random variables and the realized value of said random variable as much as possible, which is a level of clarification that is not often seen in the literature.
> The reason for this level of distinction is that we are working with random distributions (e.g., $\Delta_t$) whose realizations are themselves distributions. (The variable $\Delta_t$ could be thought of as something like a random coin among a set of biased coins, so a realization of it is a biased coin which is itself a binary probability distribution over Head and Tail. We have discussed this in detail in Section 2.1.)
> The value of $f_t$ is the realization of the random function $F_t$ which contributes to the observation as described in the first equation in Section 2.
> The function $f$ is the deterministic submodular function that is independent of time and is the expected value of $F_t$ for all $t$.
> Section 3 contains the explanation of the algorithm and that is where we explicitly mention the variable $m$.
> We have included the definition of $\tau$ in the paragraph after Lemma 1 in the revised version of the paper.
> (ii) As we mentioned in response to (1), other delay types are more informative. The composite anonymous delay, to the best of our knowledge, is the least informative type of delay in the sense that the agent receives the least amount of information with its observations.
> In non-anonymous delayed feedback, at each time-step, the agent observes a set of the form $\\{(t, r_t) \mid t \in I_t\\}$ where $I_t$ is the set of time-steps in the past whose rewards are due now. This is in sharp contrast with composite non-anonymous delayed feedback where the observation is a real number and, even if reward is a deterministic function of the action, it is often impossible to know the reward of individual actions. Therefore is it not necessarily meaningful to compare anonymous and non-anonymous delay settings.

---

### Review · Reviewer_uJf2 · 2023-09-14

**Summary Of Contributions:**

The paper studies a CMAB problem with full bandit feedback and submodular reward function. Additionally, they assume the feedback is delayed (under various delay models) and composite and anonymous, which means that all rewards from previous rounds are aggregated at the time of delivery so the learner cannot directly identify which action triggered them.
They propose an Explore-Then-Commit algorithm and analyse it in the different delay settings they consider (essentially stochastic or adversarial).

**Audience:**

Yes

**Claims And Evidence:**

Yes

**Requested Changes:**

1. Major revision of the writing, including clarification of the context and related references.

2. Add a motivation for the combination of submodular CMAB and delays.

3. Rephrase the experiment section, try other visualization techniques to highlight results and settings.

**Strengths And Weaknesses:**

### Strengths

* The paper studies a novel aspect of combinatorial bandit problems.
* The algorithm is simple and supported by an analysis in 3 different settings.
* These theoretical results are supported by experiments that include comparison to other CMAB algorithms.

### Weaknesses

* The writing needs a major revision. The introduction is unclear and does not describe related work very well. A lot of awkward formulations and incorrect English phrases are used. I give a long list of examples and suggestions for improvements in the “minor” section below.

* Related work is not very well covered. For instance, Appendix A1 cites many CMAB *without submodular rewards*, but I could find related references by a quick google search
Chen, Lixing, Jie Xu, and Zhuo Lu. "Contextual combinatorial multi-armed bandits with volatile arms and submodular reward." Advances in Neural Information Processing Systems 31 (2018).
It would be worth also saying that (Street and Golovin 2009) does address the bandit setting.

* The paper is a bit of a patchwork of existing techniques and ideas. Submodular functions are interesting, delays are interesting, but I fail to see what the combination of these two challenges makes it a more interesting problem. Can you please explain what makes this combination of problems relevant?

* The analysis descriptions in the main paper fails to convey what was the main technical challenge in combining the existing analyses. The appendix is fairly clear  but it does not highlight what part is similar to standard CMAB analysis and what part is novel, what is similar to delayed bandits literature and what is novel. If the paper required some new technical results, it would be good to highlight them.


### More detailed comments:

* The statement “As we see, ETCG outperforms almost all other baselines.” is arguable. First we don’t see much because the lines overlap a lot, and second without confidence intervals, it’s really unclear if these differences of performance are significant. For a horizon 10^5, 10 independent runs is far from enough to get meaningful confidence intervals so I am afraid the interpretation of the results cannot be that conclusive, at least not for all of the subplots.


### Questions:

“ The environment chooses a delay distribution δt ∈ T”: do you mean the delay distribution changes at every round? This does not seem standard in the stochastic delay literature. Why making this choice?


### Minor remarks:
*Introduction is awkwardly written and sometimes really unclear.*

awkward: “receives the sum total of all the rewards”

unclear: We note that in stochastic cases, the delay is not bounded, while is governed by the tight family of distributions. What does “governed” means here?

awkward again: “the process generating this delay does not need to have any independence or other assumption.” Independence of what? A process does not “have” independence.

unclear:  “measures the tightness of a family of distributions”. Tightness with respect to what? I do not understand what you mean.

awkward: “The regret bound with delayed feedback has been studied for the first time for any CMAB problem with submodular rewards in this paper, to the best of our knowledge.”

What I mean by “awkward” is that the language is imprecise or the english is incorrect. I believe a simple double check with Chat GPT or Grammarly should allow to improve the text overall, though in places where the formulation is unclear, there is a risk that Chat GPT would amplify the flaws. For instance, here is a satisfying (though not perfect) Chat GPT version of your list of contributions:
“
1. We introduce regret bounds for a stochastic CMAB problem with expected monotone and submodular rewards, a cardinality constraint, and composite anonymous feedback. Notably, we're the first to examine the delayed feedback regret bound for such CMAB problems.

2. We investigate the ETCG algorithm from Nie et al., 2022, detailing its performance in three feedback delay models: bounded adversarial delay, stochastic independent delay, and stochastic conditional independent delay.

3. Our analysis reveals the cumulative (1 − 1/e)-regret of ETCG under specific bounds for each delay model. When comparing stochastic independent and conditional independent delays, the former showcases better regret bounds. Generalizing beyond specific parameters, our findings suggest a regret bound of \(O˜(T^{2/3} + T^{1/3}ν)\) across delay models.

4. Lastly, we showcase the adaptability of our analysis for delayed feedback in combinatorial bandits, given certain algorithmic conditions. Building on Nie et al., 2023, we derive regret bounds for a meta-algorithm, highlighting its applicability to submodular bandits with knapsack constraints.
“

*Section 2*

Justification of the alpha-regret: The papers you cite to justify that alpha-regret is the right metric are referring to submodular function optimisation, not combinatorial bandits. It would be good to justify that this metric also makes sense for combinatorial bandit problems, perhaps simply with a reference in that literature, and ideally with your personal thoughts on why it’s appropriate.

“We define E_(∆t)” : Do you really need to define such notation? What is unclear with E_T(\Delta_t(x))?

Same question for “rad = \sqrt{\log(T)/m}”. I understand for the appendix but for the main paper it seems superfluous.

“Each ∆t is a random variable taking values in the set T” : wasn’t T a set of distributions??

*Section 4*

It could be good to clarify the choice of parameter m used to obtain the bounds. Did you use this optimal value in the experiments?

*Experiments*

Figure 1: “The dashed lines are y = aT2/3 for different values of a.” Why not give the corresponding two values of a?
The descriptions of the settings D1 - D6 are very ill-presented. Why not use an itemize or enumerate environment and highlight the differences to make it easily readable?

These regret curves are essentially 4 points and only differ at 10^4 and 10^5, why not use bar plots instead and show only the relevant information so we can properly see the differences you want to show?

---

> ### Author Response · Authors · 2023-10-06
>
> Thanks a lot for your detailed comments.
>
> **Weaknesses:**
>
> 1. *Related work:*
> Thank for for bringing that paper into our attention. We have included it in the related works section. However, it should be noted that the mentioned paper considers contextual setting which is more informative than the non-contextual setting which is itself more informative than delayed setting. Therefore it is only tangentially related to our work.
> We have mentioned the (Streeter and Golovin 2008) paper and we have included it in the plots, even though it considers the adversarial setting which is not a generalization of stochastic setting and therefore does not directly apply to the problem at hand.
>
> 2. *Relevance of the problem:*
> As we have mentioned in the 4th paragraph of the introduction, many real world examples of submodular combinatorial maximization problems have inherent delays. The composite anonymous delay is, to the best of our knowledge, the least informative type of delay considered in the literature. If delay is not anonymous, then the reward for each action will be precisely known at some point in the future, while this is not the case the composite anonymous setting.
> In this work, we show that the ETCG algorithm is robust under the least informative type of delay with only an additive error.
> Moreover, as we show in Section 4, the same technique could be applied to other types of combinatorial problems as well.
>
> 3. *Novelty:*
> We have updated the "contributions" segment of the introduction based on your suggestions and included a paragraph on the technical novelty as follows:
> (1) We introduce regret bounds for a stochastic CMAB problem with expected monotone and submodular rewards, a cardinality constraint, and composite anonymous feedback. Notably, this paper marks the first study of the regret bound any CMAB problem with composite delayed feedback, including CMAB with submodular rewards.
> (2) We investigate the ETCG algorithm from (Nie et al., 2022),, detailing its performance in three feedback delay models: bounded adversarial delay, stochastic independent delay, and stochastic conditional independent delay.
> Specifically, this is the first study where the distribution of stochastic delay is permitted to vary over time.
> This introduces novel models for stochastic delayed composite anonymous feedback, which are more general than those previously explored in existing literature.
> (3) Our analysis reveals the cumulative $(1 - 1/e)$-regret of ETCG under specific bounds for each delay model. When comparing stochastic independent and conditional independent delays, the former showcases better regret bounds. Generalizing beyond specific parameters, our findings suggest a regret bound of $\tilde{O}(T^{2/3} + T^{1/3}\nu)$ across delay models.
> (4) Lastly, we showcase the adaptability of our analysis for delayed feedback in combinatorial bandits, given certain algorithmic conditions.
> Building on (Nie et al., 2022), we derive regret bounds for a meta-algorithm, highlighting its applicability to other CMAB problems such as submodular bandits with knapsack constraints (See Nie et al., 2023).
> On the technical side, we define new generalized notions of delay and introduce the notion of upper tail bound, which measures the tightness of a family of distributions.
> As discussed in Appendix A.2, algorithms designed for composite anonymous feedback, including those in our study, rely on the concept of repeating actions a sufficient number of times to minimize the impact of delay on the observed reward.
> We employ Bernstein's inequality to control the effect of previous actions on the observed reward of the current action that is being repeated.
> This approach enables us to establish an upper bound on regret, expressed in terms of the expected value of the upper tail bound.
>
> **More detailed comments:**
>
> We have improved the plots with error bars and added experiments for the horizon $10^6$.
>
> **Questions:**
>
> That is correct.
> As we have mentioned above (in response to Weaknesses-3), the delay distribution is allowed to change over time.
> We have included the following example in the revised version to make this more clear.
>
> To elaborate on the nature of the delay, let us ignore the combinatorial aspect of the problem for the moment and consider the following setting.
> A retailer, that sells both food and computer products, can buy an advertisement slot on an E-commerce platform, e.g., Amazon or eBay.
> This is a 2-armed bandit where we assume that the retailer buys an ad slot for a product at each time-step.
> We further assume that each time-step is a single day and the only information revealed to the retailer every day is the total added revenue as a result of the advertisements.
>
> (continued in the next comment)

---

> ### Author Response · Authors · 2023-10-06
>
> A delay distribution is a sequence of real numbers that add to one, e.g., $\delta = (0.9, 0.05, 0.05, 0, \cdots)$.
> Such a delay means that 90% of the reward (increase in revenue as a result of the ads) is received immediately, while 5% of the reward is received in each of the next 2 time-steps.
> Clearly it is not enough to consider a fixed delay distribution.
> Therefore we consider a situation where $\Delta$ is a random variable where $\delta$ is a realization of $\Delta$.
>
> It is reasonable to assume that the effect of an ad for food is more immediately seen in the revenue compared to the effect of an ad for computer products.
> Therefore we may consider a setting where $\Delta_{F}$ is a random delay distribution corresponding to food and $\Delta_{C}$ correspond to computer products and $\Delta_F \neq \Delta_C$.
> This corresponds to the setting considered in (Wang et al.,2021) and (Garg \& Akash, 2019).
>
> Now assume that a sale for computer products, but not food, is going to start next week.
> Modeling this scenario means that $\Delta$ should change over time, but should also depend on the action, since only one of the actions is affected by the sales.
> This is *Unbounded Stochastic Conditionally Independent Delay* considered in our paper.
>
> If we instead assume that the delay changes over time, but does not depend on the arm (for example if the retailer is selling different types of computer products), then this will be *Unbounded Stochastic Independent Delay*.
>
> Finally, if delay is too complicated to be covered by previous settings, then we consider *Bounded Adversarial Delay*.
> For example, consider a scenario where different retailers pay the E-commerce platform for advertisement slots, but when the ad is shown depends on the buyers and the actions of other retailers, which can not be known in advance. The boundedness assumption guarantees that for each ad slot purchased, the effect on the revenue of the retailer will be limited to a fixed time, e.g. one month, from the purchase of the ad.
>
> **Minor remarks:**
>
> We note that some of the awkwardness in the introduction is the result of the generality of the setting we consider where we have to postpone some details to the main body.
> We have rewritten several of the phrases you mentioned, including the list of contributions, and added extra descriptions or footnotes.
> The following comments require separate responses which we include here.
>
> - *unclear: “measures the tightness of a family of distributions”. Tightness with respect to what? I do not understand what you mean.*
> The concept of "tightness of family of distributions" is similar to the concept of "boundedness of a sequence of numbers", in the sense that we do not need to specify boundedness of a sequence of numbers with respect to something.
> In this analogy, Lemma 1 corresponds to the statement "a sequence of real numbers $(a_i)_{i = 0}^\infty$ is bounded if and only if there is a number $M$ such that $a_i \leq M$ for all $i \geq 0$".
> The intuitive idea of tightness is that the family of distributions do not have a non-zero probability mass "escaping to infinity".
> The Wikipedia page for "Tightness of measures" describes this notion in a more general setting.
> We have included a definition of tightness after Assumption 1, where we first discuss this notion in the main text. In our revised version, we have included a reference to Assumption 1 in the introduction.
> - *“We define $E_{\mathcal{T}}(\Delta_t)$” : Do you really need to define such notation? What is unclear with $E_{\mathcal{T}}(\Delta_t(x))$?*
> Yes. Note that $\Delta_t$ is a random distribution and each realization of it is a distribution which has an expected value. As we have discussed in Section 2.1, the expression $E_{\mathcal{T}}(\Delta_t)$ is a distribution and denotes the average of $\Delta_t$ with respect to the first randomness, but not the second. On the other hand, the $\Delta_t(x)$ is a random variable taking its values in $[0,1]$ and $E_{\mathcal{T}}(\Delta_t(x))$ is a real number in $[0,1]$. More precisely, the values of the sequence $(E_{\mathcal{T}}(\Delta_t(0)), E_{\mathcal{T}}(\Delta_t(1)), E_{\mathcal{T}}(\Delta_t(2)), ...)$ add up to one and therefore can be thought of as a probability distribution, namely $E_{\mathcal{T}}(\Delta_t)$, over the set of non-negative integers.
> - *"Each $\Delta_t$ is a random variable taking values in the set $\mathcal{T}$" : wasn’t $\mathcal{T}$ a set of distributions?*
> That is correct. Please refer to the previous response and our response to the "Questions" section above.
> - *It could be good to clarify the choice of parameter $m$ used to obtain the bounds. Did you use this optimal value in the experiments?*
> We have mentioned the choice of $m$ in the algorithm which is the optimal value and is the only value that was used in the experiments. We have also updated our submission to mentioned the choice of $m$ in the main text.

---

> ### Author Response · Authors · 2023-10-14
>
> Please note that we have updated the description of experiments and added Appendix I where we have included experiments showing the added regret when we move from a setting without delay to a setting that has delay.

---

### Review · Reviewer_A6rz · 2023-10-09

**Summary Of Contributions:**

This paper studies combinatorial submodular bandits with delayed composite rewards. They consider various assumptions on the delays, stochastic independent, correlated with the rewards and adversarial. For each model, they show that the regret scales with O(T^{2/3}+d T^{1/3}) where d is some parameter concerning the delays (e.g. expected delay or maximal delay). This is achieved by an algorithm ETCG which seems to be the same as in Nie et al (2022), with no modifications to account for the delayed feedback. In addition to the theoretical results, the authors show that experimentally the ETCG algorithm is more robust to delayed feedback.

**Audience:**

Yes

**Claims And Evidence:**

Yes

**Requested Changes:**

- Either prove that the obtained rate is optimal or show that a modified algorithm can achieve improved regret, or at the very least justify why these things cannot be done.
- clarify experiments section
- demonstrate experimentally that the delay dependence scales with $T^{1/3}$ as indicated by the theoretical results.

**Strengths And Weaknesses:**

The authors obtain the same order of regret bounds for stochastic and adversarial generated delays through a unified analysis that just leaves one term to be analyzed separately in the different settings. This is mathematically pleasing, however I wonder if it is potentially missing something as in other delayed feedback settings we can do significantly better when the delays are iid stochastic (typically an additive term independent of the horizon) compared to adversarial. In particular, I wonder whether the results for the stochastic delays are tight?

The experimental results are interesting and demonstrate that the ETCG algorithm performs better than others. Interestingly all algorithms in the experiments are not tailored for delayed feedback. This suggests that the authors interest is in investigating the robustness of the algorithms to delayed feedback in this form of bandit problems, whereas I read the introduction as intending to develop algorithms which perform best with delayed feedback. I also found the experimental setups quite difficult to understand and identify from the plots. Importantly, it wasn’t clear to me that the experimental results were directly validating the theoretical ones. In particular, it would be interesting to include an experiment showing that indeed the effect of delayed feedback does indeed increase at a rate of $T^{1/3}$. This wasn’t clear to me from the experiments, and it would be good to validate the theoretical results and draw more connections between the two.

I also do not understand the motivation for not making any adjustments to the ETCG algorithm and simply analysing the effects of delayed feedback on the vanilla algorithm. The ETCG algorithm is a phase based algorithm, which are known to perform well under delayed feedback, but to get good performance in the presence of delays it is typically required to adjust the phase lengths to account for the delayed feedback. For example, consider the stochastic setting where we know that with high probability, each delay is bounded by UCB(\tau). Then, if the phase lengths are extended by UCB(\tau) then we can be confident that we will have received as much information in a phase of the delayed algorithm as in the non-delayed setting. This means that the rest of the analysis should go through smoothly, with the only difference being that we incur an extra UCB(\tau) regret per batch. If the number of batches are small, this could lead to a delay penalty which is smaller than the T^{1/3}d term that appears in the present analysis. Therefore, I wonder if there is a relatively straightforward way of adapting the algorithm to improve the regret and address one of the main limitations of the paper. Perhaps I have missed a difficulty which makes this impossible, but I would appreciate if the authors could justify not extending the phase lengths and explain if doing so would make any difference to performane.

---

> ### Author Response · Authors · 2023-10-14
>
> Thanks a lot for your comments and for appreciating our results.
>
> The problem of aggregated anonymous delay was first studied by (Pike-Burke et al., 2018). This notion was generalized to composite anonymous delay in (Cesa-Bianchi et al., 2018) in the adversarial feedback setting. Later (Garg \& Akash, 2019) and (Wang et al., 2021) studied the same delay type in the stochastic feedback setting. Our work is the study of the same type of delay for combinatorial bandits in general and combinatorial submodular bandits in particular.
> Many different types of delay has been studied in the literature but to the best of our knowledge, the composite anonymous delay is the least informative type.
> We note in all of these works, including ours, a key idea is to repeat actions enough times so that we can extract meaningful information. This is not always necessary in other types of delay. In particular, if delay is not anonymous, there is no need to repeat actions since we will eventually know the reward for each action.
> Other algorithms discussed here for combinatoral bandits do not repeat actions and therefore there is little hope of them achieving desirable results in the presence of composite anonymous delay.
> Moreover, it should be noted that other algorithms do not address the general problem of stochastic submodular combinatoral bandit and we have simply included them to have something to compare to.
> As described in Appendix H, other baselines are either for adversarial setting (e.g. OG$^o$), which is different than stochastic, or have stronger assumptions (e.g. CMAB-SM and DART), or are not designed for combinatoral setting (e.g. ARS-UCB). Thus, we note that ETCG is the only algorithm with regret guarantees for submodular bandits with full-bandit feedback in the absence of delay. This is the key reason for only analyzing this algorithm in the presence of delay.
>
> The idea you mentioned about adjusting the length of different phases is implemented in ARS-UCB algorithm of (Wang et al., 2021) which we have included in our experiments. The issue when moving from MAB to combinatorial MAB is that we can not guarantee revisiting any action after the end of the phase. In fact, even trying each action once would result in exponential time and space complexity. For example, for N = 100 and K = 40, if trying each action takes a nanosecond, it would take ~ 30 times the age of the universe to try each action once. Therefore we can not directly use any UCB type algorithm. In the exploration phase of the ETCG algorithm, each action is only taken within at most one phase. This makes it difficult, if not impossible, to use an idea similar to ARS-UCB and change the length of phases to improve the regret bounds.
>
> As for the lower bounds on regret, we note that the optimal regret bound for stochastic submodular maximization with bandit feedback is not known even when there is no delay present.
> We have updated the description of experiments and added Appendix I where we have included experiments showing the added regret when we move from a setting without delay to a setting that has delay.

---

### Author Response · Authors · 2025-01-30

The revised version of this paper is accepted at IEEE Transactions on Artificial Intelligence. The readers are referred to https://ieeexplore.ieee.org/abstract/document/10835119 for an updated version of this work.

---

### Decision · Action_Editor_Ur3y · 2024-01-26

**Recommendation:** Reject

**Comment:**

As described above, the paper in its current form is very preliminary, and neither the ideas nor their execution warrant publication. The recommended improvement in the algorithm design and more extensive experiments (analyzing what factors determine the behavior of the algorithms) could be considered for future improvements.

**Audience:**

During the discussion concluding the review process, the reviewers found that the contribution is a straightforward combination of existing results, and this, combined with the very niche setting considered in the paper would severely limit the potential audience of the paper. Analyzing an already existing algorithm which is not designed for the examined setting, the paper does not shed much light on the intricacies of this problem apart from the known fact that phase-based algorithms work well for delayed composite anonymous feedback (independently from minor deviations in the problem settings). While the reviewers offered some suggestions on how to modify the algorithm to customize it better to the considered setting (e.g., changing the phase length), this was simply dismissed by the authors, although it could easily lead to some immediate improvements.

Considering all this, the potential audience for the paper is extremely limited.

**Claims And Evidence:**

The paper considers the stochastic submodular bandit problem with delayed composite anonymous feedback. An existing algorithm is analyzed for 3 different delay models theoretically and experiments are also presented.

While the theoretical claims seem correct, the experiments are very simplistic and not fully analyzed, contributing very little to the understanding of the problem (e.g., for small sample sizes the suggested algorithm does not outperform the baselines, and it is not explored how this depends on the setting, such as the number of arms relative to the length of the horizon or the difference of the rewards of the different actions; also, the new experiments analyzing the extra cost of the delay show a lot of unexpected variations). The reviewers also found the description of the experiments somewhat hard to understand (e.g., if the confidence intervals are computed for the same problem or they correspond to the random selection of the problem instances), and some suggestions (e.g., repeating the experiments more than 10 times) are not followed.

It is not clearly argued how the motivating examples would be a good fit to the suggested setting and what technical difficulties arise in the analysis.